# The Surprising Agreement Between Convex Optimization Theory and Learning-Rate Scheduling for Large Model Training

**Fabian Schaipp** [1]  **Alexander Hägele** [2]  **Adrien Taylor** [1]  **Umut Şimşekli** [1]  **Francis Bach** [1]

## Abstract

We show that learning-rate schedules for large model training behave surprisingly similar to a performance bound from non-smooth convex optimization theory. We provide a bound for the constant schedule with linear cooldown; in particular, the practical benefit of cooldown is reflected in the bound due to the absence of logarithmic terms. Further, we show that this surprisingly close match between optimization theory and practice can be exploited for learning-rate tuning: we achieve noticeable improvements for training 124M and 210M `Llama`-type models by (i) extending the schedule for continued training with optimal learning-rate, and (ii) transferring the optimal learning-rate across schedules.

## 1. Introduction

Large-scale machine learning requires a fine-tuned training recipe. In particular, the choice of appropriate learning-rate schedules is a crucial step for classical optimization methods. This usually decomposes into the choice of a *schedule*, determining the shape of learning rates over time, and the tuning of a multiplicative *base learning-rate*, determining the magnitude of the step sizes.

Over the years, the `cosine` schedule (Loshchilov & Hutter, 2017) has emerged among the most commonly used schedules in large (language) model training (Brown et al., 2020; Touvron et al., 2023). The standard practice is to set the frequency of the cosine to half of the total number of training steps (Hoffmann et al., 2022); as a consequence, the entire schedule depends on the length of training, which makes it unsuitable for continued training. Recently, it has been shown that the performance of `cosine` can be matched

[1]Inria, Departement d'Informatique de l'Ecole Normale Superieure, PSL Research University, Paris, France [2]EPFL, Lausanne, Switzerland. Correspondence to: Fabian Schaipp <fabian.schaipp@inria.fr>.

*Proceedings of the $42^{nd}$ International Conference on Machine Learning*, Vancouver, Canada. PMLR 267, 2025. Copyright 2025 by the author(s).

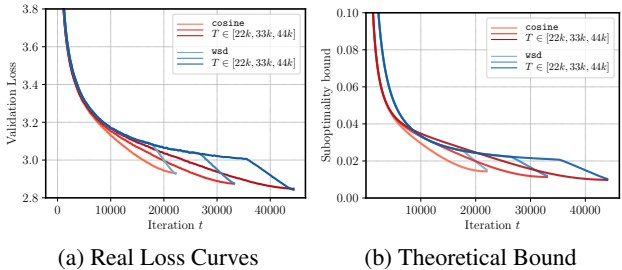

|  |  |
|---|---|
| (a) Real Loss Curves | (b) Theoretical Bound |

*Figure 1.* Strikingly similar: Validation loss for a 210M `Llama` model trained with `AdamW` **(left)** and the theoretical suboptimality bound (6) from convex optimization **(right)**. Both plots show `wsd` and `cosine` schedule with different training lengths $T$, and with base learning-rate of `cosine` being twice as large as for `wsd`.

by an arguably much simpler schedule, that combines a constant part with a cooldown period in the end (Hu et al., 2024; Hägele et al., 2024). This alternative schedule is established under the name `wsd` (*warmup-stable-decay*). One distinguishing feature of `wsd` is the drastic decrease of the loss shortly after initiating the cooldown.

However, the recent advancements in learning-rate scheduling have emerged almost exclusively from *empirical* rather than from *theoretical* considerations (Loshchilov & Hutter, 2017; Goyal et al., 2017; Hoffmann et al., 2022; Hägele et al., 2024). We do not yet have a fundamental understanding that could explain the features of the above-mentioned schedules and why they perform better or worse at a given task, restraining the tuning procedure to a trial-and-error approach.

**Summary and contributions.** In this paper, we show that several empirical findings on scheduling can be reproduced with a suboptimality bound for `SGD` on convex problems that was introduced by Defazio et al. (2023). Among others, this theory allows to reproduce (i) the matching performance of `cosine` and `wsd`; (ii) the necessity and optimal length of the cooldown period in the end of training (see Section 4).

In a second step, we take the reverse direction and show how the theoretical bound can be exploited in practice (see Section 5): for continued training of a 124M and 210M `Llama`-style model, using the theoretically optimal schedule

notably improves performance compared to continuing with the same learning rate; it also allows to transfer the optimal learning-rate across schedules (Figs. 10 and 12). This leads us to the perhaps surprising conclusion that the empirical behavior of learning-rate schedules in (non-convex) deep learning can be described precisely with a theoretical bound from non-smooth stochastic convex optimization.

A particular focus of this paper is put on the wsd schedule: we derive a convergence result for this schedule (without warmup) in the non-smooth stochastic convex setting, see Section 3.2. Most importantly, the cooldown period of wsd leads to vanishing log-terms in the bound, which provides an explanation of the benefit of cooldown observed in practice. Second, we show that the sudden drop during cooldown can be observed in *upper* and *lower* bounds of the suboptimality, as well as for a non-smooth convex toy problem. Code for all experiments is available at https://github.com/fabian-sp/lr-scheduling.

**Setup.** We consider the training problem

$$\min_{x \in \mathbb{R}^d} f(x), \quad f(x) := \mathbb{E}_s[f(x, s)]. \quad (1)$$

In the above, $x \in \mathbb{R}^d$ are the learnable parameters of a machine learning model, and $f$ is a loss function. The expectation is taken over the distribution of a random variable $s$ that maps to the space or set $\mathcal{S}$ (typically the training set). We assume that $f(\cdot, s)$ has a suitable subdifferential for every $s \in \mathcal{S}$ (for example, see Rockafellar (1970); Clarke (1983); Bolte & Pauwels (2021)). We denote elements of the subdifferential as $g \in \partial f(x, s)$.[1]

We study the iterates of SGD with a learning-rate schedule, given by

$$x_{t+1} = x_t - \gamma \eta_t g_t, \quad g_t \in \partial f(x_t, s_t), \quad t \in \mathbb{N}. \quad (2)$$

Here, $\gamma > 0$ is called the *base learning-rate* and $\eta_t > 0$ is called the *schedule*. While it might seem redundant to separate $\gamma$ from $(\eta_t)$, this reflects the standard practice in deep learning libraries such as Pytorch. Most importantly, for different schedules (constant, cosine, wsd,...), the optimal value of $\gamma$ is in general different.

We remark that the most commonly used optimizer for training in practice is Adam(W) (Kingma & Ba, 2015; Loshchilov & Hutter, 2019), and all empirical results we present or refer to in this paper are obtained with Adam(W). However, the theoretical results apply to SGD; we address this limitation in detail in Section 6.

Cosine **and** wsd **schedules.** We now formally introduce the two running examples cosine and wsd. Without

warmup, the wsd schedule is constant up to iteration $T_0 \leq T$, then decays linearly to zero. Formally, we have

$$\eta_t = \begin{cases} 1 & 1 \leq t < T_0, \\ 1 - \frac{t-T_0}{T+1-T_0} & T_0 \leq t \leq T+1. \end{cases} \quad (3)$$

The cosine schedule is given by $\eta_t = \frac{1}{2}(1 + \cos(\frac{t-1}{T}\pi))$ for $1 \leq t \leq T+1$. Note that for both schedules we have $\eta_{T+1} = 0$ (we choose $T+1$ in order to ensure that $\eta_t > 0$ for $t \leq T$). It is also common to decay the cosine to a factor of $0.1$ of the peak learning-rate instead of $0$.

**Notation and naming convention.** We will use wsd in the paper as it is the most established abbreviation in the literature; however, similar to Zhai et al. (2022); Hägele et al. (2024), we will refer to the phase where the schedule decays to zero as *cooldown* instead of *decay*, in order to avoid confusion with other terminology (e.g., *weight decay*). Unless explicitly stated otherwise, $\|\cdot\|$ and $\langle\cdot,\cdot\rangle$ denote the standard Euclidean norm and its scalar product.

## 2. Related Work

**Learning-rate schedules.** The cosine schedule (Loshchilov & Hutter, 2017) can be considered the de-facto default in large-scale deep learning. Convergence results for SGD with cosine schedule have been shown by Li et al. (2021). Recently, the wsd schedule (short for *warmup-stable-decay*, also called trapezoidal schedule) has been proposed as an alternative (Zhai et al., 2022; Hu et al., 2024; Hägele et al., 2024). Hägele et al. (2024) show that wsd matches the performance of cosine on LLM pretraining, while largely reducing the compute needed for scaling-law experiments, as the constant part of the schedule can be reused.

**Last-iterate convergence.** We will see that it is crucial to use a bound for the *last-iterate* in order to closely match empirical loss curves. This is in contrast to many standard convergence results that prove an upper bound on the quantity $\min_{t=1,\ldots,T} \mathbb{E}[f(x_t) - f(x_\star)]$. Due to convexity and Jensen's inequality, the same bound usually holds for $\mathbb{E}[f(\bar{x}_T) - f(x_\star)]$, where $\bar{x}_T$ is some (weighted) average over $\{x_1,\ldots,x_T\}$. Last-iterate results, that is, bounds on $\mathbb{E}[f(x_T) - f(x_\star)]$, are less standard: convergence of SGD has been proven for constant step sizes (Zhang, 2004), and for decreasing step sizes in bounded domains (Shamir & Zhang, 2013). Other results are restricted to a specific choice of schedule (Jain et al., 2021; Zamani & Glineur, 2023). The backbone of this article will be a result from Defazio et al. (2023), which proves a last-iterate bound for general schedules; compared to previous work (Orabona, 2020) it has the advantage that the bound remains meaningful if the last step size $\eta_T$ is very small.

---

[1]In case the reader is uncomfortable with the notion of subdifferentials, the entire article can be read with $g_t$ being the gradient $\nabla f(x_t, s_t)$ instead.

**Understanding cooldown.** For the wsd schedule, one can consistently observe a sudden drop in train/validation loss after the start of the cooldown phase (Hägele et al., 2024). Hu et al. (2024) find that the curvature of the loss increases during cooldown; Hägele et al. (2024) expand this and conclude that "the cooldown phase is a smooth transition to a basin in the loss landscape". More recently, Wen et al. (2025) hypothesize that the sudden drop is caused by a *river-valley loss landscape*, that arises from "heterogeneity in the stochasticity of different tokens". In this work, we will offer an additional (and potentially much simpler) model: the drop of the loss can be observed in upper and lower bounds of the suboptimality, based on first-order convex optimization theory. In particular, this phenomenon happens for a toy instance of $\ell_\infty$-norm regression.

## 3. Convergence Results

Let us assume convexity of the objective and recall the definition of the iterates.

(A1) For each $s \in \mathcal{S}$ the function $f(\cdot, s) : \mathbb{R}^d \to \mathbb{R}$ is convex, that is, for all $x, y \in \mathbb{R}^d$ and $g \in \partial f(x, s)$

$$f(y, s) - f(x, s) \geq \langle g, y - x \rangle. \tag{4}$$

(A2) Let $\gamma > 0$ and $\eta_t > 0$. For $t \in \mathbb{N}$, consider the iterates

$$x_{t+1} = x_t - \gamma \eta_t g_t, \quad g_t \in \partial f(x_t, s_t). \tag{5}$$

Let $x_\star \in \mathbb{R}^d$ be an arbitrary point of interest, for example the (local) minimum of $f$ that is closest to $x_1$. We do not make any other assumption on $x_\star$ for now.

**Theorem 3.1** (cf. Thm. 10 from Defazio et al. (2023)). *Let $(x_t)$ be given by (A2), with $\eta_t > 0$ for $t = 1, \ldots, T$ and $\gamma > 0$. Let $x_\star \in \mathbb{R}^d$ and define $D := \|x_1 - x_\star\|$ and $\bar{\eta}_T := \sum_{t=1}^T \eta_t$. Under (A1), for any $T \in \mathbb{N}$ it holds*

$$\mathbb{E}[f(x_T) - f(x_\star)] \leq \frac{1}{2\gamma \bar{\eta}_T} \Big[ D^2 + \gamma^2 \sum_{t=1}^T \eta_t^2 \mathbb{E}\|g_t\|^2 \Big]$$

$$+ \frac{\gamma}{2} \sum_{k=1}^{T-1} \frac{\eta_k}{\sum_{t=k+1}^T \eta_t} \Big( \frac{1}{\sum_{t=k}^T \eta_t} \sum_{t=k}^T \eta_t^2 \mathbb{E}\|g_t\|^2 \Big). \tag{6}$$

The above result is essentially the same as (Defazio et al., 2023, Thm. 10); the only difference is that we explicitly separate $\gamma$ and $(\eta_t)$ which will be convenient subsequently. We refer to Appendix E for a proof.

Our next goal is to compute the base learning-rate $\gamma$, given a schedule $\eta_t$, that minimizes the bound in (6). To do so, we assume a bound on the expected gradient norms:

(A3) Assume that there exists $(G_t)_{t \leq T} > 0$ such that $\mathbb{E}\|g_t\|^2 \leq G_t^2$ for all $t \leq T$.

*Remark* 3.2. In general, the choice of $\gamma$ will affect the iterates $(x_t)$ and therefore the gradient norm bounds $(G_t)$. Thus, the following Corollary can be applied *only if* we apply the same bound $G_t$ independent of $\gamma$. This is the case for the standard assumption of $f(\cdot, s)$ being Lipschitz with constant $G > 0$; in that case, choose $G_t = G$ for all $t \in \mathbb{N}$.

Let $\eta_{1:T} := (\eta_1, \ldots, \eta_T)$, and $G_{1:T} := (G_1, \ldots, G_T)$. For convenience, we define the quantities

$$\mathcal{T}_1(\eta_{1:T}, D, T) := \frac{1}{2\sum_{t=1}^T \eta_t} D^2,$$

$$\mathcal{T}_2(\eta_{1:T}, G_{1:T}, T) := \frac{1}{2\sum_{t=1}^T \eta_t} \Big( \sum_{t=1}^T \eta_t^2 G_t^2 \Big) \tag{7}$$

$$+ \frac{1}{2} \sum_{k=1}^{T-1} \frac{\eta_k}{\sum_{t=k+1}^T \eta_t} \Big( \frac{1}{\sum_{t=k}^T \eta_t} \sum_{t=k}^T \eta_t^2 G_t^2 \Big).$$

**Corollary 3.3.** *In the setting of Theorem 3.1, under (A3), for any $T \in \mathbb{N}$ it holds $\mathbb{E}[f(x_T) - f(x_\star)] \leq \Omega_T$ with*

$$\Omega_T := \frac{\mathcal{T}_1(\eta_{1:T}, D, T)}{\gamma} + \gamma \mathcal{T}_2(\eta_{1:T}, G_{1:T}, T). \tag{8}$$

*For a given $(G_t)$, $D$ and $T$, minimizing the right-hand side of (8) with respect to $\gamma > 0$ gives the solution $\gamma^\star = \sqrt{\frac{\mathcal{T}_1(\eta_{1:T}, D, T)}{\mathcal{T}_2(\eta_{1:T}, G_{1:T}, T)}}$. Plugging $\gamma^\star$ back into (8), we have $\mathbb{E}[f(x_T) - f(x_\star)] \leq 2\sqrt{\mathcal{T}_1(\eta_{1:T}, D, T)\mathcal{T}_2(\eta_{1:T}, G_{1:T}, T)}$.*

Next, we plug in the cosine and wsd schedule into Theorem 3.1. Applying[2] Corollary 3.3 with $T \to t$, we get $\mathbb{E}[f(x_t) - f(x_\star)] \leq \Omega_t$ for $t \in [T]$ with

$$\Omega_t := \frac{\mathcal{T}_1(\eta_{1:t}, D, t)}{\gamma} + \gamma \mathcal{T}_2(\eta_{1:t}, G_{1:t}, t). \tag{9}$$

### 3.1. Comparison of cosine and wsd

For a training horizon $T \in \mathbb{N}$, we define both schedules $(\eta_t)_{1 \leq t \leq T+1}$ such that they reach $\eta_{T+1} = 0$. For a formal definition of wsd and cosine see (3) and thereafter. For each different training horizon $T$, and for both schedulers, we pick the optimal base learning-rate $\gamma^\star$ given by Corollary 3.3 and plot the bound $\Omega_t$ in Fig. 2 (with $G_t = D = 1$ for all $t \in \mathbb{N}$). We plot a sweep of $\gamma$ in Fig. 3a.

Perhaps surprisingly, the shape of the theoretical bound $\Omega_t$ (for the convex case) matches closely the empirical loss curves of (the non-convex problem of) language model pretraining in Hägele et al. (2024); see Fig. 1 for a side-by-side comparison. This is especially visible in the sudden drop of the loss for the wsd schedule during cooldown. However, using the last-iterate result is crucial for this: we demonstrate

---

[2] For Corollary 3.3 we require $\eta_t > 0$ for $t = 1, \ldots, T$, which is why we construct the schedule such that $\eta_{T+1} = 0$ instead of $\eta_T = 0$.

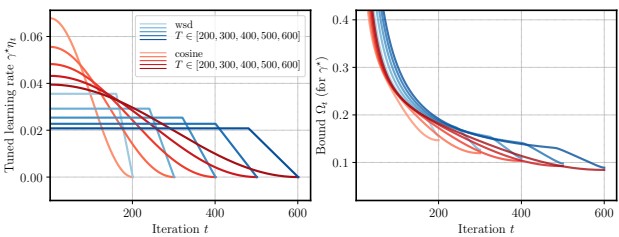

*Figure 2.* Schedule **(left)** and theoretical bound **(right)** for `cosine` and `wsd`, and various $T$, with base learning-rate $\gamma^\star$.

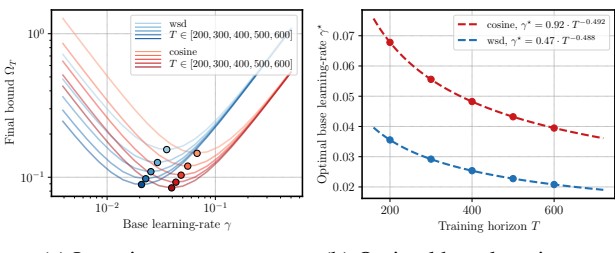

(a) Learning-rate sweep      (b) Optimal base learning-rate

*Figure 3.* Optimal base learning-rate decays with inverse square-root of training horizon $T$ **(right)**. For `cosine`, it is roughly twice as large as for `wsd` (as $0.92/0.47 \approx 2$). The dashed curve in the right-hand side plot is obtained with a least-squares fit.

this with an ablation study that uses a standard bound on the minimum suboptimality instead; there, the theoretical bound does not resemble empirical loss curves (cf. Appendix A).

> **Takeaway:** The last-iterate bound in Corollary 3.3 matches the shape of the loss curves in Hägele et al. (2024). In particular, the sudden drop for `wsd` during cooldown can be observed.

> **Takeaway:** The optimal base learning-rate from Corollary 3.3 scales $1/\sqrt{T}$ with the training horizon, and is roughly twice as large for `cosine` as for `wsd` (Fig. 3b). This matches empirical observations (Fig. 4 in Shen et al. (2024) and Fig. 3 in Hägele et al. (2024)).

### 3.2. Bound for `wsd` Schedule

We now derive the bound in Corollary 3.3 for $(\eta_t)$ being the `wsd` schedule. To the best of our knowledge, this schedule has not been analyzed theoretically before. For this section, assume that $G_t = G > 0$ for all $t \in \mathbb{N}$. A useful notation will be the *harmonic number $H_t$*, defined as $H_t := \sum_{k=1}^{t} \frac{1}{k}$ for $t \in \mathbb{N}$, and $H_0 := 0$. We recall that $H_t$ behaves like $\ln(t)$ in the limit. As baseline, we first compute the bound for the constant schedule.

**Constant schedule.** If $\eta_t = 1$ for all $t \in \mathbb{N}$, it is easy to compute $\mathcal{T}_1(\eta_{1:T}, D, T) = \frac{D^2}{2T}$, $\mathcal{T}_2(\eta_{1:T}, G_{1:T}, T) =$

$\frac{G^2}{2}[1 + H_{T-1}]$. Therefore, Corollary 3.3 yields

$$\mathbb{E}[f(x_T) - f(x_\star)] \leq \frac{DG}{\sqrt{T}}\sqrt{1 + H_{T-1}}.$$

**The `wsd` schedule.** We will now compute a suboptimality bound for the `wsd` schedule (without warmup). We will show that if higher-order terms are ignored, the improvement of `wsd` over a constant schedule is essentially due to the absence of the logarithmic term $H_{T-1}$. In Theorem 3.4, we surpress some terms due to space constraints. The full version is given in Theorem G.1.

**Theorem 3.4.** *Assume that (A3) holds with $G_t = G$ for some $G > 0$ for all $t \in \mathbb{N}$. Let $\gamma = \gamma^\star$ from Corollary 3.3. Then, for the `wsd` schedule (3) with $1 \leq T_0 < T$ we get*

$$\mathbb{E}[f(x_T) - f(x_\star)] \leq DG\sqrt{\frac{4}{T+T_0}\left[\Lambda_1 + \Lambda_2 - \Lambda_3 + o(1)\right]}$$

*where $\Lambda_1 := \frac{2}{3} + \frac{T+2T_0}{3(T+T_0)}$, $\Lambda_2 := H_{T+T_0-2} - H_{T-T_0+1}$, $\Lambda_3 := \frac{(T-T_0)(T_0-1)}{3(T-T_0+2)(T+T_0)}$ and $o(1)$ summarizes terms that go to zero as $T \to +\infty$.*

Assume that the cooldown length is proportional to $T$, that is, $T_0 = \beta T$ for $\beta \in (0,1)$. For large $T$, we have $\Lambda_3 \approx \frac{(1-\beta)\beta T^2}{3(1-\beta)(1+\beta)T^2} = \frac{\beta}{3(1+\beta)}$. Using Lemma D.3, we can estimate $H_{(1+\beta)T-2} \leq 1 + \ln((1 + \beta)T)$ and $H_{(1-\beta)T+1} \geq \ln((1-\beta)T)$. This yields $\Lambda_2 \leq 1 + \ln(\frac{1+\beta}{1-\beta})$. Altogether,

$$\mathbb{E}[f(x_T) - f(x_\star)] \lessapprox \frac{DG}{\sqrt{T}}\sqrt{\frac{4}{1+\beta}\left[\frac{5}{3} + \Lambda_4 + \ln(\frac{1+\beta}{1-\beta})\right]},$$

where $\Lambda_4 := \frac{1+2\beta}{3(1+\beta)} - \frac{\beta}{3(1+\beta)}$. In total, the term in the square-root **does not contain logarithmic terms in** $T$. This is the main difference to the constant schedule (where in the square-root we have $1 + H_{T-1} \approx 1 + \ln(T)$). See Fig. 20 for a visualization. We defer additional remarks and the proof of Theorem 3.4 to Appendix G.

> **Takeaway:** The `wsd` schedule improves over the constant schedule by a logarithmic term. This improvement in the bound happens during the cooldown period (cf. Figs. 2 and 20).

## 4. Theoretical Simulations

In the following, we simulate the bound from Theorem 3.1 in order to analyze its dependence on the cooldown length for `wsd`, and on the gradient norm magnitude. Additional experiments (e.g., on the `cosine` cycle length, and a comparison of classical schedules) and supplementary information are deferred to Appendix B. Unless explicitly mentioned, we set $G_t = 1$ for all $t \in N$ and $D = 1$ for the entire simulation. We do not use warmup for neither schedule.

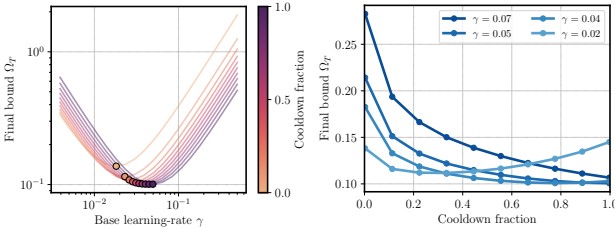

*Figure 4.* **(Left)** Optimal base learning-rate increases with cooldown fraction. **(Right)** For fixed $\gamma$, the optimal cooldown fraction can be smaller than 1. The analogous curves for real experiments with similar parabola shapes are in Fig. 21.

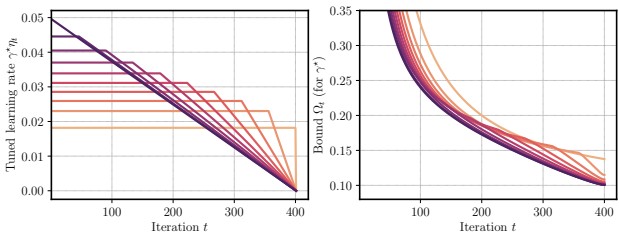

*Figure 5.* Schedule **(left)** and theoretical convergence **(right)** for varying cooldown fraction. With optimal base learning-rate $\gamma^\star$, starting the cooldown at $T_0 = 1$ is optimal. Fig. 21 shows the analogous plot for real experiments with the same behavior.

### 4.1. Cooldown Length

Previously, we have set $T_0 = 0.8 \cdot T$ for `wsd`. In Figs. 4 and 5, we vary the *cooldown fraction*, defined as $\frac{T-T_0}{T}$. Specifically, we vary from $T_0 = T$ to $T_0 = 1$ (constant schedule to linear-decay schedule similar to Defazio et al. (2023, Corollary 2)).

> **Takeaway:** The simulation suggests that *if the base learning-rate $\gamma$ is fully tuned, then the optimal cooldown fraction is 1 (linear decay). For fixed $\gamma$, the optimal cooldown fraction can be smaller than one.*

The first observation is in line with empirical observations from Defazio et al. (2023) that compares many different schedules across several machine learning tasks, and find that the linear-decay schedule performs best on average. Further, it is known that the linear-decay schedule matches the exact lower-bound convergence bound for the (stochastic) convex, Lipschitz case (Defazio et al., 2023; Zamani & Glineur, 2023); see Appendix G for detailed comments. The second observation matches the finding of Hägele et al. (2024): in Appendix B.6, Fig. 21 we show the analogous figure on real training data (also see Fig. 5 in Hägele et al. (2024)). For small base learning-rate $\gamma$, we obtain the same parabola shape; however, for large enough $\gamma$, the parabola turns into a monotonically decreasing curve.

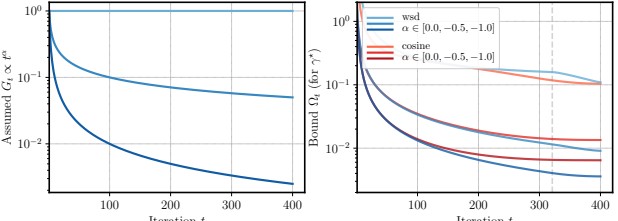

*Figure 6.* Assumed gradient shape **(left)** and theoretical convergence **(right)**. Only with $\alpha = 0$ (constant $G_t$), the sudden drop for `wsd` is clearly visible.

### 4.2. Gradient Norm

We now analyze how the bound of the expected gradient norms $G_{1:T}$ influences the shape of $\Omega_t$. In this section only, we assume that $G_t = t^\alpha$, $\alpha \in \{0, -0.5, -1\}$. We sweep the base learning-rate $\gamma$ by computing the minimal $\Omega_T$ from (9) for the above choice of $G_{1:T} = (G_1, \ldots, G_T)$. We set $T = 400$, and the cooldown fraction to $0.2$ for `wsd`. Fig. 6 shows that the sudden drop in loss for `wsd` is only visible if $G_t$ does not go to zero as $t \to \infty$.

> **Takeaway:** The sudden drop during cooldown is most pronounced if the gradient norms do not go to zero.

Interestingly, if the gradient norms go to zero, the `wsd` schedule also obtains a significantly better bound than `cosine`. So far we have observed that non-vanishing gradient norms lead to the characteristic drop in the upper bound $\Omega_t$. Next, we show that the same phenomenon can be observed for (i) a suboptimality lower bound and (ii) for the loss of the iterates of `SGD` on a simple non-smooth convex problem.

### 4.3. Lower Bounds and Convexity

In all previous sections we analyzed an upper bound $\Omega_t$ of $\mathbb{E}[f(x_t) - f(x_\star)]$. How tight is this upper bound? To answer this, we compute lower bounds of $\mathbb{E}[f(x_t) - f(x_\star)]$ using the PEP framework: for a given function class and algorithm, a worst-case example can be constructed by solving a semidefinite program (Drori & Teboulle, 2014; Taylor et al., 2017a;b; Goujaud et al., 2024). Additional details are provided in Appendix B.4.

> **Takeaway:** The sudden drop during cooldown appears as well for the PEP lower bound (Fig. 7). The worst-case suboptimality value at $T = 60$ is very similar for `cosine` and `wsd`.

In Section 4.2, we have shown that non-vanishing gradient norms are characteristic for the sudden drop (of the upper bound $\Omega_t$) during cooldown of `wsd`. We observe the same behavior for the *actual loss* when running gradient descent with the `wsd` schedule for the 2-dimensional, convex, non-

smooth problem $\min_{x \in \mathbb{R}^2} \|Ax - b\|_\infty$. The experimental details and plots are deferred to Appendix B.1.

> **Takeaway:** The sudden drop of the loss for `wsd` is not specific to model architecture or even to non-convexity. It can be observed when minimizing a simple non-smooth, convex objective function (Fig. 17).

# 5. Applications

So far, we have shown that the bound from Theorem 3.1 matches very closely empirical loss curves. However, the bound in Theorem 3.1 contains quantities that are unknown in practice, such as the gradient norm bounds $G_t$ and $D$. Thus, the question arises how to convert the theoretical result into practical applications. The following two scenarios demonstrate that using the optimal schedule and base learning-rate predicted from theory improves pretraining of a 124/210M `Llama`-style transformer (Vaswani et al., 2017; Touvron et al., 2023).

## 5.1. Schedule Construction for Continued Training

The first application is to construct learning-rate schedules for longer horizons: for example, assume we have trained a model for $T_1$ steps, but later want to continue training up to $T_2 > T_1$ steps. The main benefit of the `wsd` schedule is that the training steps up to the cooldown phase can be reused, thus reducing the amount of additional compute required for continual training (Hägele et al., 2024). This is not true neither for the linear-decay nor for the `cosine` schedule, as the value of the schedule in each step depends on the total number of steps.

Assume we have tuned the base learning-rate $\gamma^\star$ of `wsd` for the short run $T_1$. We have seen in Fig. 3b that $\gamma^\star$ decreases with $T$; thus, continuing training at $\gamma^\star$ until $T_2$ would use a suboptimal learning rate. We present two solutions:

(B1) We have seen that $\gamma^\star$ *increases* with the cooldown fraction $c$ (Section 4.1). We can increase the cooldown fraction $c_1$ for the long training run to $T_2$, to compensate for the decrease in $\gamma^\star$ due to $T_1 \to T_2$.

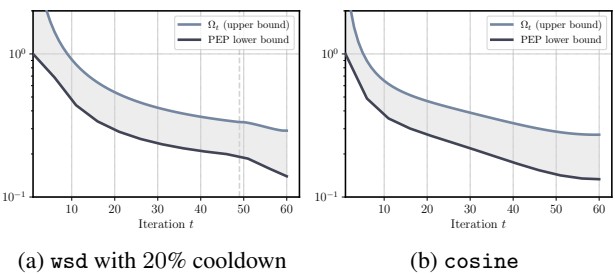

(a) `wsd` with 20% cooldown     (b) `cosine`

*Figure 7.* PEP lower bound matches the upper bound $\Omega_t$ in shape.

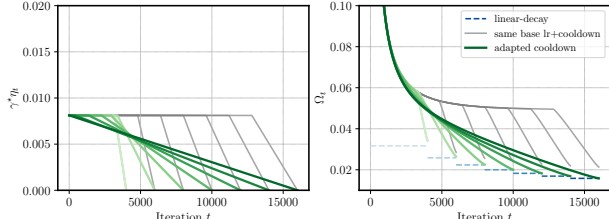

*Figure 8.* **(Left)** Transferring the `wsd` schedule from horizon $T_1 = 4000$ to $T_2 \in [1.5T_1, 4T_1]$. **(Right)** Not adapting the cooldown length leads to significant suboptimality. Dashed horizontal lines mark bound for the linear-decay schedule with tuned $\gamma^\star$.

(B2) Alternatively, we can keep the same cooldown fraction, but decrease the learning-rate in the steps from $T_1$ to $T_2$: assume a piecewise constant schedule with $\eta_t = 1$ for $t$ up to the start of cooldown of the short run, and $\eta_t = \rho$ for $t$ up to the start of cooldown of the long run. How do we need to set $\rho$, such that $\gamma^\star$ remains the optimal base-learning rate for this schedule?

**Simulation.** We simulate both options in Figs. 8 and 9. Here, we set $T_1 = 4000$ and $T_2$ ranging from $1.5T_1$ to $4T_1$. We construct the extended schedule by sweeping $c_1$ for (B1) and $\rho$ for (B2), and picking the value where the optimal base learning-rate according to Corollary 3.3 is closest to $\gamma^\star$. The cooldown phase of the short run is set to 20%. Specifically, our analysis suggests to decrease the schedule by $\rho = 0.525$ for $T_2 = 2T_1$ and by $\rho = 0.375$ for $T_2 = 4T_1$ (see Fig. 22, left). We verified that changing the values of $G$, $D$, or $T_1$ do not affect the result (plots not shown); the values might be different for other cooldown fractions than 20%.

For (B2) (Fig. 9), we conclude that by decreasing the schedule by the correct factor $\rho$, we can reuse the entire constant part of the short run, while obtaining a bound $\Omega_t$ close to the bound for a tuned linear-decay schedule. Importantly, keeping the same base learning-rate for the entire long run would result in a significantly worse bound $\Omega_t$. For (B1) (Fig. 8), the required increase in cooldown fraction is large, and hence for long extensions, only small parts of training can be reused. When doubling the training length ($T_2 = 2T_1$), the adapted cooldown fraction is roughly $c_1 = 0.6$. As an alternative, one could use the `1/sqrt` schedule, defined by $\eta_t := 1/\sqrt{t}$, combined with cooldown (Zhai et al., 2022). Fig. 23 shows that for `1/sqrt` the cooldown fraction can roughly stay the same, which however comes at the cost of a larger gap to linear-decay. From a theoretical and practical perspective, we conclude that the approach (B2) is preferable, as it allows to reuse the entire short run with no drawbacks in terms of the bound, and – in a similar fashion as before – allows iteratively continuing from the newly obtained checkpoints of a constant learning-rate phase.

**Experiments.** Based on the above, we extend the training of a 124M and 210M Llama-style transformer (Touvron et al., 2023) on the SlimPajama dataset (Soboleva et al., 2023). For details on model, dataset and training procedure see Appendix B.5. We set $T_1 = 50$k and $T_2 \in \{100\text{k}, 200\text{k}\}$; a sweep over 50k steps gives $\gamma^\star_{50\text{k}} \approx 0.001$. As a baseline, we use a wsd schedule that continues with the same $\gamma$ over the extended training length. For the adapted schedule from (B2), we decrease the learning-rate after 40k steps linearly over 1000 steps (e.g., from $10^{-3}$ to $5.25 \cdot 10^{-4}$) as a precautionary measure; however, we did not observe that a decrease in-one-go results in significantly different performance. We use a cooldown of 20% for all runs.

Considering the results in Fig. 10, we conclude that the schedule adaptation suggested by theory leads to a slight but noticeable improvement in validation loss for both extended horizons. The improvement is more pronounced for the larger 210M model. Moreover, we observe a sudden drop in loss after decreasing the schedule at 40k steps, analogous to what the theoretical bound predicts, albeit the loss decrease thereafter is slower than expected (cf. Fig. 9). We also test adapting the cooldown length as described in (B1): for a total length of 100k steps, if cooldown is initialized immediately after 40k steps (cooldown 60%), we observe even larger improvements as previously (see Fig. 24).

From Figs. 10 and 24, we see that the improvement in loss of the adapted wsd schedule over a naive continuation is in the range of 0.01. This raises the natural question of the relevance of such an improvement. To answer this, we estimate the slope of our loss curves[3]: we find that for $T_2 = 100$k, a decrease of 0.01 takes roughly 6k steps in the constant learning-rate phase; for $T_2 = 200$k, it takes roughly 14.5k steps. This translates to 0.6B and 1.5B tokens, respectively. Notably, to match the adapted wsd schedule, this would require a substantial amount of 6% and 7.25% longer training. Another way to reason about the significance of the loss improvement is through the use of *scaling laws*, which leads to very similar estimates (see Appendix B.5).

### 5.2. Learning-Rate Transfer Across Schedules

One insight from Corollary 3.3 is that if $G_t = G$, then the dependence of $\gamma^\star$ on $G$ and $D$ is multiplicative. This implies that if we know $\gamma^\star$ for a given practical problem, any multiplicative transfer can be realized. For example, assume we know the optimal base learning-rate for the wsd schedule with cooldown fraction $c \in [0, 1]$, and let us denote the tuned value as $\gamma^\star(c)$. As we have seen, the linear-decay schedule ($c = 1$) attains the optimal bound; thus, to obtain a

---

[3]We do linear regression on the loss values of the baseline run between $[64\,000, 84\,000]$ for the 100k run, and between $[144\,000, 164\,000]$ for the 200k run. This accounts proportionally for the cooldown.

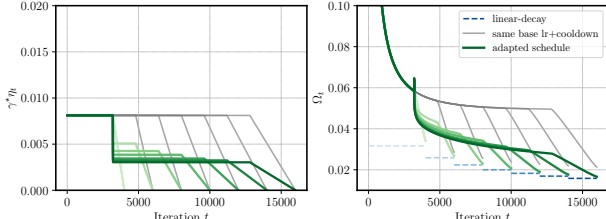

*Figure 9.* Transferring the learning-rate schedule from horizon $T_1 = 4000$ to $T_2 \in [1.5T_1, 4T_1]$ (see also Fig. 22, left). Decreasing the learning rate **(green)** after the short run (at iteration 3200) leads to significant better bound $\Omega_t$ as keeping it constant **(grey)**. Dashed horizontal lines **(blue)** mark bounds for linear-decay schedule with tuned $\gamma^\star$.

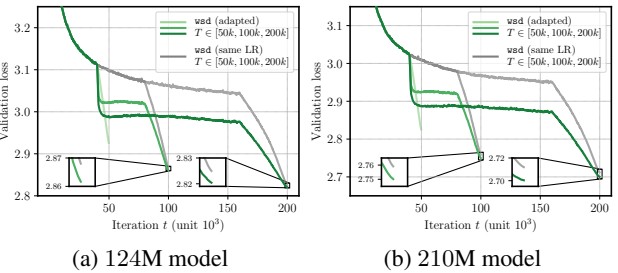

(a) 124M model      (b) 210M model

*Figure 10.* Transferring the learning-rate schedule from horizon $T_1 = 50\,000$ to $T_2 \in [2T_1, 4T_1]$. Decreasing the base learning-rate **(green)** after 40k steps leads to small improvements in validation loss compared to keeping it the same **(grey)**. We discuss the significance of the difference in loss values of (around 0.01) in Section 5.1 and Appendix B.5. See Fig. 22 for schedules.

---

better model, we might want to retrain with the linear-decay schedule.[4] However, we do not yet know $\gamma^\star(1)$. Can we compute $\gamma^\star(1)$ from $\gamma^\star(c)$ based on the theoretical bound?

**Simulation.** In Fig. 11 we show the quantity $\ln\left(\frac{\gamma^\star(1)}{\gamma^\star(c)}\right)$ for $c \in (0, 1)$. We simulate both the linear cooldown (3), and the 1-sqrt cooldown which has the form $\eta_t = 1 - \sqrt{\frac{t - T_0}{T + 1 - T_0}}$ (Hägele et al., 2024). Across several orders of $T$, the results are consistent; for example, knowing $\gamma^\star$ for 20 % of linear cooldown, we can compute $\gamma^\star(1) \approx e^{0.7}\gamma^\star(0.2)$. For Fig. 11, we set $G = D = 1$; the resulting curve looks the same if we vary $D$ or $G$ (plots not shown).

**Experiments.** We now analyze the quantity $\ln\left(\frac{\gamma^\star(1)}{\gamma^\star(c)}\right)$ with real data (training a 124M Llama-style model for 50k steps), with linear cooldown. We estimate $\gamma^\star(c)$ from a grid of base learning-rates $\gamma$ and cooldown fractions $c$ (see Fig. 12b and Appendix B.5 for details on this step). We plot

---

[4]For example, assume that $\gamma^\star(c)$ has been made public on Github or we obtained it from a sweep that used the wsd schedule due to practical constraints.

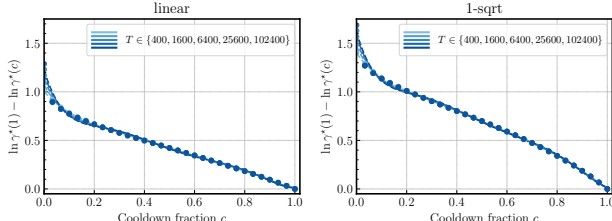

*Figure 11.* Transferring the optimal base learning-rate from cooldown fraction $c$ to linear-decay ($c = 1$): for linear cooldown **(left)** and `1-sqrt` cooldown **(right)**. Dashed lines are fitted polynomial of degree 6.

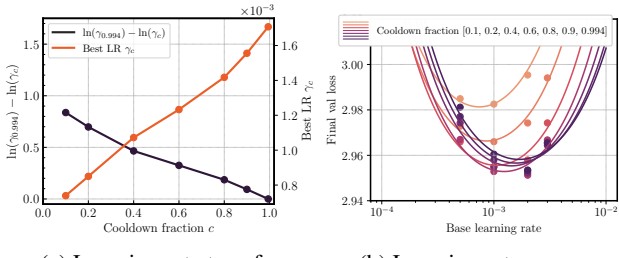

(a) Learning rate transfer      (b) Learning rate sweep

*Figure 12.* **(Left)** Re-analysis of learning-rate transfer (Fig. 11) for 124M model. $\gamma_c$ denotes the best performing base learning-rate for cooldown fraction $c$, estimated from a sweep **(right)**. We observe that the learning-rate transfer (black line) almost perfectly matches the predictions by theory (e.g., $\gamma(0.994) \approx e^{0.7}\gamma^\star(0.2)$). Note that the maximal cooldown fraction is 0.994 due to warmup and corresponds to a full linear schedule.

$\ln(\frac{\gamma^\star(1)}{\gamma^\star(c)})$ in Fig. 12a; it matches almost perfectly with the curve predicted from theory in Fig. 11. This implies that knowing the optimal base learning-rate for 20% cooldown, one can immediately transfer the learning-rate to linear-decay (100% cooldown), *without any additional sweeps*; for the setup we consider, the linear-decay run obtains a final validation loss of 2.9535 vs. the best run with 20% cooldown obtaining a final loss of 2.9660.

## 6. Limitations

We have shown that the empirical performance of various learning-rate schedules for large model training reflects closely the theoretical suboptimality for non-smooth stochastic convex optimization. We want to stress that we can not expect the bound from Theorem 3.1 to match training curves *perfectly*: first, it is an upper bound of the loss for convex problems only, and in practice many other factors (e.g., randomness, architecture choices, data mixture) and training techniques (e.g., loss function, weight decay) will impact convergence and stability of training (Wortsman et al., 2024).

The perhaps most glaring limitation of our work is that it

is based on a theoretical result for SGD, while most of the empirical evidence we use is obtained with Adam(W). More generally, the result in Theorem 3.1 can not explain any performance differences that stem from the optimization algorithm. However, we believe that this gap can be closed in future work for several reasons: (i) by showing similar theoretical results for the methods used in practice; as a first step, we provide a proof for mirror descent (an entire family of methods) in Appendix F. It has been shown that for diagonal networks, the iterates of SGD are equivalent to mirror descent on a convex problem formulation (Even et al., 2023). (ii) Several recent variants of SGD close the gap to Adam on transformer problems (Kunstner et al., 2023; Xu et al., 2024). (iii) It has been shown that most of the parameters of language models can be equally well trained with SGD (Zhao et al., 2025).

The second obvious limitation of Theorem 3.1 is the convexity assumption, while modern deep learning problems are non-convex. At this point we have no explanation for why the convex theory is still closely matching (some) real-world observations. However, it has been shown that the landscape of neural network optimization problems might be reasonably close to being convex (Hardt et al., 2018; Liu et al., 2023; Islamov et al., 2024). The sudden performance increase during cooldown is not restricted to language modeling and has also been reported for image problems, e.g., training ResNets with SGD (Sandler et al., 2023) or ViTs (Zhai et al., 2022). We verify this through additional experiments for SGD on Imagenet in Appendix C, which further contains experiments on OpenWebText2.

Finally, the empirical quantity we compare to the theoretical bound is the test loss. This is limited to situations where the generalization gap between training and test loss is negligible; that being said, the current practice of single-pass training for large models falls within this category (Aitchison, 2024; Xiao, 2024).

## 7. Conclusion

In this paper, we show that learning-rate schedules in practice behave surprisingly similar to what convex optimization theory predicts. This spans across the necessity and optimal length of the cooldown period at the end of training as well as the optimal learning-rate transfer. Notably, our experiments suggest that the theoretical bounds can be used as *testbed for schedule design* before training: we have shown that theoretically inspired schedules achieve notable improvements in practical scenarios. More broadly, our results suggest that one key characteristic underlying the observed behavior is gradient norms that do not go to zero; in practice, this could be due to non-smoothness (of the objective) or due to the problem-inherent gradient noise. We leave it as future work to explain this phenomenon.

## Acknowledgments

A. Taylor is supported by the European Union (ERC grant CASPER 101162889). U. Şimşekli is supported by the European Union (ERC grant DYNASTY 101039676). Views and opinions expressed are however those of the author(s) only and do not necessarily reflect those of the European Union or the European Research Council Executive Agency (ERCEA). Neither the European Union nor the granting authority can be held responsible for them. The French government partly funded this work under the management of Agence Nationale de la Recherche as part of the "France 2030" program, reference ANR-23-IACL-0008 (PR[AI]RIE-PSAI).

## Impact Statement

This paper presents work whose goal is to advance the field of Machine Learning. There are many potential societal consequences of our work, none which we feel must be specifically highlighted here.

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

# Appendix

The supplementary material is organized as follows:

## A. Ablation: Min-Suboptimality Bounds

Standard convergence results for the SGD method (2) make statements on the suboptimality of an average iterate, or of the best objective value (in expectation) found up to iteration $T$. We state one of the standard results for the non-smooth (stochastic) convex setting below (Zinkevich, 2003):

**Theorem A.1.** *Assume that each $f(\cdot, s)$ is convex. Let $(x_t)$ be the iterates given by (A2), with $\eta_t > 0$ and $\gamma > 0$. Let $x_\star \in \mathbb{R}^d$ and define $D := \|x_1 - x_\star\|$. Under (A3), we have*

$$\min_{t=1,\dots,T} \mathbb{E}[f(x_t) - f(x_\star)] \leq \frac{1}{2\gamma \sum_{t=1}^{T} \eta_t}\Big[D^2 + \gamma^2 \sum_{t=1}^{T} \eta_t^2 G_t^2\Big]. \tag{10}$$

*The right-hand side of the above is minimized by $\gamma^\star = \frac{D}{\sqrt{\sum_{t=1}^{T} G_t^2 \eta_t^2}}$. Plugging in $\gamma^\star$ yields*

$$\min_{t=1,\dots,T} \mathbb{E}[f(x_t) - f(x_\star)] \leq \frac{D\sqrt{\sum_{t=1}^{T} G_t^2 \eta_t^2}}{\sum_{t=1}^{T} \eta_t}.$$

We now repeat the theoretical simulations, but, instead of $\Omega_t$ from (9), using

$$\Omega_t = \frac{1}{2\gamma \sum_{s=1}^{t} \eta_s}\Big[D^2 + \gamma^2 \sum_{s=1}^{t} \eta_s^2 G_s^2\Big] \tag{11}$$

### A.1. Ablation of Section 3.1

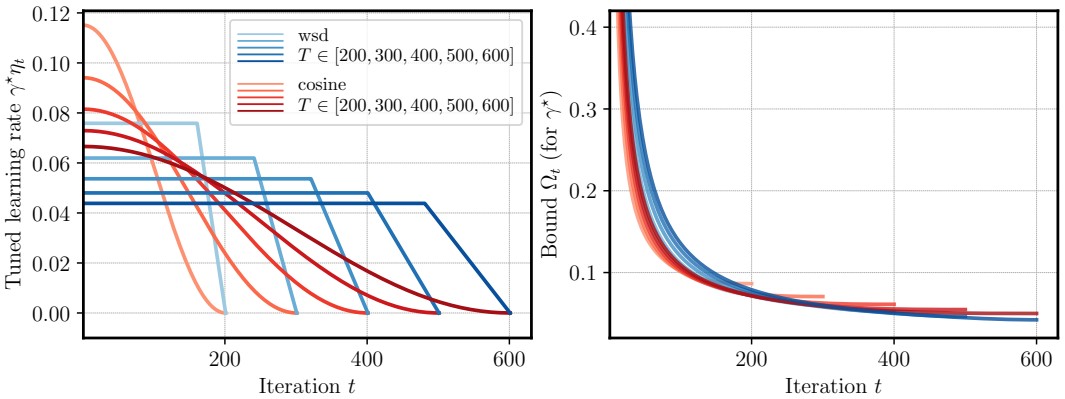

*Figure 13.* Same as Fig. 2, but with $\Omega_t$ from (11)

The bound on the best-so-far bound has a very different shape of the last-iterate bound. This shows that standard bounds such as in Theorem A.1 do not capture the real-world convergence observed in Hägele et al. (2024).

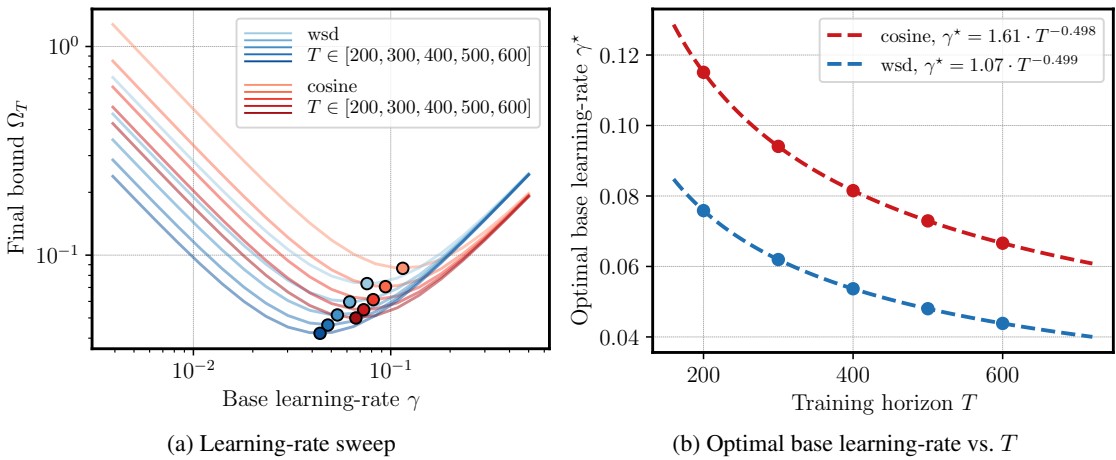

(a) Learning-rate sweep

(b) Optimal base learning-rate vs. $T$

*Figure 14.* Same as Fig. 3, but with $\Omega_t$ from (11)

## A.2. Ablation of Section 4.1

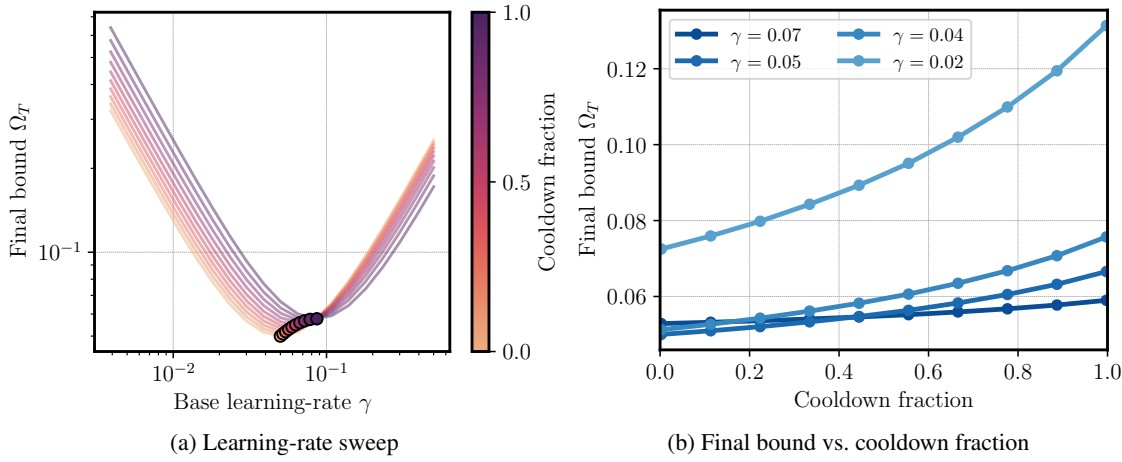

(a) Learning-rate sweep

(b) Final bound vs. cooldown fraction

*Figure 15.* Same as Fig. 4, but with $\Omega_t$ from (11)

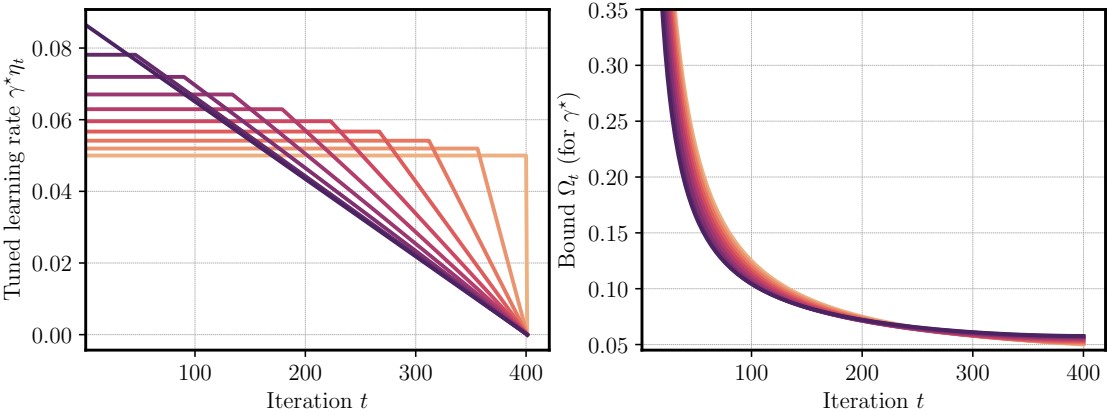

*Figure 16.* Same as Fig. 5, but with $\Omega_t$ from (11)

# B. Experiments: Supplementary Material

## B.1. Non-smooth Convex Example

Here, we provide details for the non-smooth convex toy example of $\min_{x \in \mathbb{R}^d} \|Ax - b\|_\infty$ mentioned in Section 4.3. We set $d = 2$ and pick $A \in \mathbb{R}^{20 \times d}$ uniformly at random from $[-1, 1]$. We generate an oracle $x_\star \in \mathbb{R}^d$ and set $b = Ax_\star$. We then run gradient descent (GD) for $T = 400$ iterations with the wsd schedule (cooldown fraction 0.2 and $\gamma = 0.02$). As baseline, we plot the constant schedule with $\gamma = 0.02$ and a cosine schedule with $\gamma = 0.04$. We pick zero as starting point, except for the constant schedule, where we pick $(10^{-3}, 10^{-3})$ to obtain a visually distinguishable path.

The objective function and iterate paths are shown in Fig. 17.

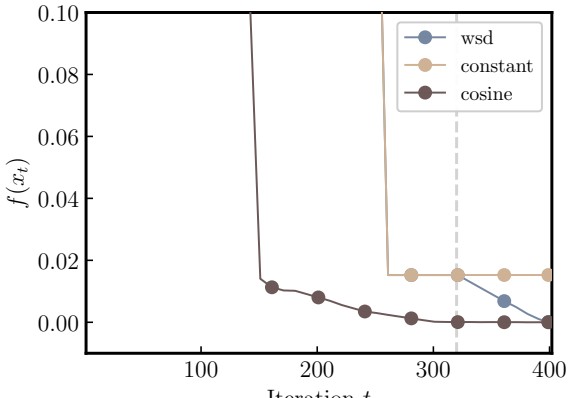 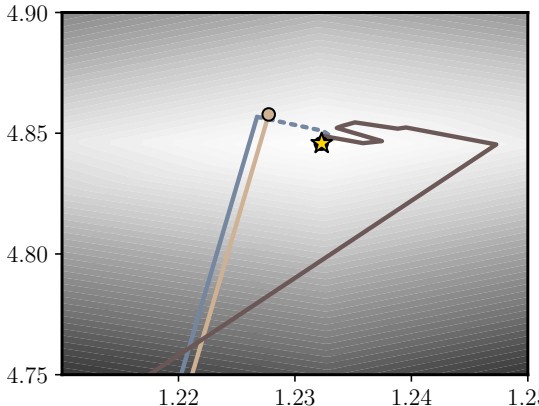

*Figure 17.* (**Left**) Sudden drop of the loss for wsd schedule for a convex, non-smooth problem. (**Right**) Iterate path for the three schedules. For wsd, the cooldown period is indicated with the dashed line. Star marks solution.

## B.2. Schedule Comparison

We compare the upper bound $\Omega_t$ from (9) for various schedules:

- wsd with cooldown fraction 0.2,

- cosine,

- constant schedule,

- linear-decay schedule, that is, wsd with cooldown fraction of 1,

- 1/sqrt schedule, where $\eta_t = 1/\sqrt{t}$,

- 1-sqrt schedule, where $\eta_t = 1 - \sqrt{\frac{t-1}{T}}$.

We assume $D = 1, G_t = 1$ and set $T = 400$. For each schedule we sweep the base learning-rate $\gamma$ and plot the bound $\Omega_t$ for $\gamma = \gamma^\star$ obtained from Corollary 3.3.

## B.3. Cosine Cycle Length

For the cosine schedule, an important hyperparameter is its *cycle length*, that is, the amount of training where the schedule first reaches zero. Originally, it was proposed in Loshchilov & Hutter (2017) to use multiple warm restarts (a cycle length less than one). Later, Hoffmann et al. (2022) show empirically that the best performance in language modeling tasks is obtained by setting the cycle length to one (the half-cosine matches exactly the training duration).

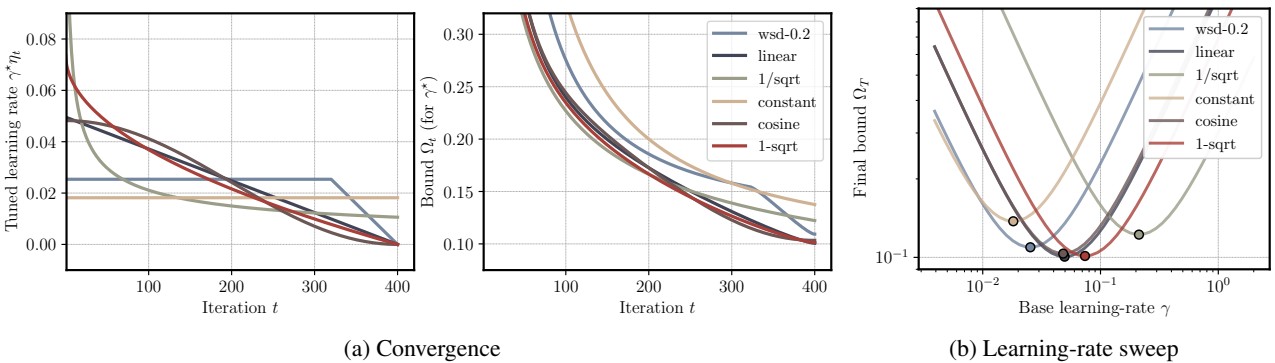

(a) Convergence
(b) Learning-rate sweep

*Figure 18.* Comparison of various learning-rate schedules. Convergence is plotted with the optimal base learning-rate $\gamma^\star$ (chosen individually for each schedule).

To the best of our knowledge, this recommendation is based mostly on empirical insights. Using the bound obtained in Theorem 3.1, our analysis shows that a cycle length of one obtains the lowest bound $\Omega_T$. Thus, the theoretical bound is in accordance to the empirical conclusion from Hoffmann et al. (2022).

Note that Hoffmann et al. (2022) choose the base learning-rate $\gamma$ equally for all cycle lengths. To match the setting of their experiment, we pick $\gamma^\star$ for a cycle length of one, and use this for all other cycle lengths as well. Picking $\gamma^\star$ for each cycle length individually yields qualitatively the same result (the optimal cycle length being one), but with slightly less pronounced differences (plots not shown). In contrast to previous simulations, the final value of the schedule is chosen as 0.1 of the peak learning-rate (instead of zero), again in order to match the setting of Hoffmann et al. (2022).

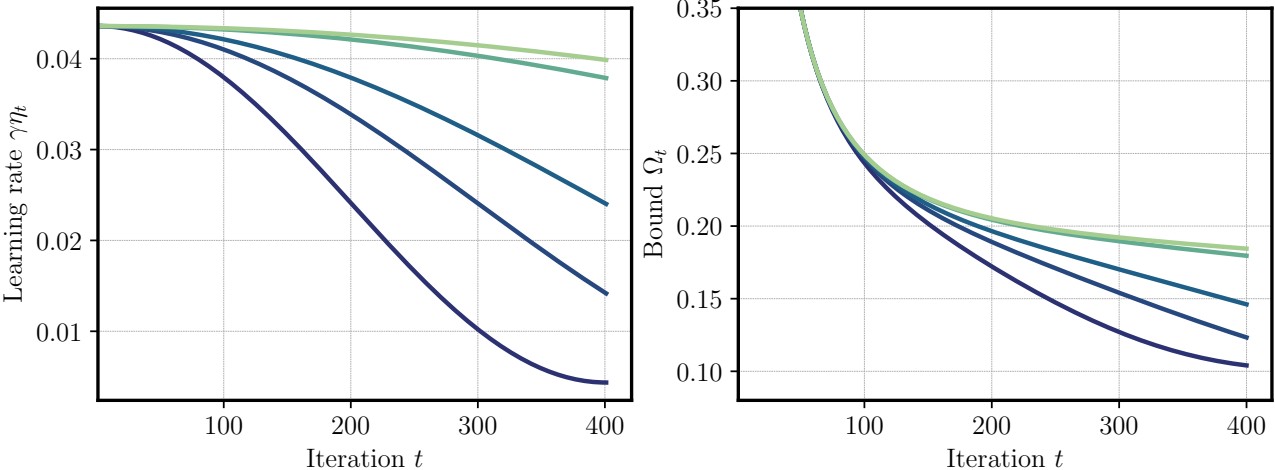

*Figure 19.* Comparison of cycle lengths for the `cosine` schedule. Compare to Figure A1 in Hoffmann et al. (2022).

### B.4. Details on Lower Bound Computation

We provide additional details for the simulation in Section 4.3. We compute the lower bounds with the `PEPit` package (Goujaud et al., 2024). For our purpose, we use the class of convex $G$-Lipschitz functions and gradient descent (`GD`) with the step sizes $\gamma \cdot \eta_t$. In `PEPit`, we use the `MOSEK` solver. As the size of the semidefinite program grows with $T$, we choose a rather small $T = 60$, and compute the lower bound of $\mathbb{E}[f(x_t) - f(x_\star)]$ for all $t + 1$ that are multiple of 5. Note that the specific worst-case function $f$ constructed by `PEPit` can be different for each $t$ (as it maximizes the suboptimality exactly at iteration $t$). We set $\gamma$ to $\gamma^\star$ minimizing the upper bound $\Omega_t$ (cf. Corollary 3.3). We set $G = D = 1$.

## B.5. Details on Experiments in Fig. 1a and Section 5

**Training details.** The loss curves in Fig. 1a are an exact reproduction of the curves in (Hägele et al., 2024, Fig. 3); they are obtained from training a 210M `Llama`-style transformer (Touvron et al., 2023). The base learning-rate of `cosine` is 0.001, and for `wsd` it is 0.0005.

All of the following applies to the training runs used in the experiments in Section 5: we use exactly the same model architecture as in Hägele et al. (2024), which is a `Llama`-style transformer with 12 (24) layers and 12 attention heads for the 124M (210M) model. The dataset used for training is `SlimPajama` (Soboleva et al., 2023). Specifically, for runs with 50 000 steps (5B tokens), we use the `SlimPajama-6B` subset available on Hugging Face (link below). For the extension runs with 100 000 and 200 000 steps (approximately 10B and 20B tokens), we randomly sample 550M documents (roughly 5% of full corpus) from the full `SlimPajama-627B` to arrive at a corpus of 30B tokens.

We train for 50 000 steps, where the first 300 steps are reserved for linear warmup. We use `AdamW` (Loshchilov & Hutter, 2019) with a weight decay of 0.1. For all further details we refer to Hägele et al. (2024, App. A.1). Note that all training curves show the validation loss computed over a subset of 32 batches, while the final validation loss is computed over approx. 6 000 batches; hence, the final value of the loss curve might not be identical to the final loss computed over the full validation set. One single run over 50 000 steps takes roughly 2 hours on two Nvidia H100 GPUs.

The training runs can be reproduced with the following repositories:

Training code from Hägele et al. (2024): `https://github.com/epfml/schedules-and-scaling/`
Dataset: `https://huggingface.co/datasets/DKYoon/SlimPajama-6B`

**Fitting procedure.** We execute training runs on a grid of base learning-rates $\gamma \in \{0.0005, 0.001, 0.002, 0.003\}$ and cooldown fractions $c \in \{0.1, 0.2, 0.4, 0.6, 0.8, 0.9, 0.994\}$. Note that the largest cooldown fraction is slightly smaller than 1 as the remaining 0.6% percent of steps constitute warmup. The final validation set loss (after 50k steps) for all runs is displayed in Fig. 12b (every dot marks one single training run).

We then fit, for each cooldown fraction $c$ separately, a function of the form $h_c(\gamma) = \frac{A_c}{\gamma} + B_c\gamma + C_c$, where $A_c, B_c, C_c$ are fittable parameters. The resulting function is plotted as solid line in Fig. 12b. The functional form of $h_c(\gamma)$ is inspired by the bound (8).

We then approximate the optimal base learning-rate $\gamma^\star(c)$ by computing the minimizer of $h_c(\gamma)$. The result of this step is plotted in red in Fig. 12a.

**Assessing loss differences through scaling laws.** In this section, we estimate with scaling laws how much more parameters or training data/steps would be needed to make up a loss difference of 0.01 (see end of Section 5.1 for context). The Chinchilla law (Hoffmann et al., 2022) states that the loss $L(N, D)$ for a model with parameters $N$ after training for $D$ tokens can be estimated with

$$L(N, D) = E + \frac{A}{N^\alpha} + \frac{B}{D^\beta} \, , \tag{12}$$

where $E, A, B, \alpha, \beta$ are usually fitted from data. More concretely, assume we have trained a model of size $N_1$ for $D_1$ tokens. To arrive at an improvement of $\delta$ with a new combination of the number of parameters and tokens to $(N_2, D_2)$, we obtain

$$\delta = L(N_1, D_1) - L(N_2, D_2) = A\left(\frac{1}{N_1^\alpha} - \frac{1}{N_2^\alpha}\right) + B\left(\frac{1}{D_1^\beta} - \frac{1}{D_2^\beta}\right)$$

Consequently, we can consider two cases:

- Case 1: Fix $N_1 = N_2$. That is, we fix a parameter size and look for the number of tokens by which we need to extend the training to improve the loss by $\delta$. Solving the above equation then gives

$$D_2 = \left(\frac{1}{D_1^\beta} - \frac{\delta}{B}\right)^{-\frac{1}{\beta}} \, .$$

- Case 2: Fix $D_1 = D_2$. This is the case where we estimate the size that would achieve the desired loss improvement for the same training data. Similarly, this results in

$$N_2 = \left( \frac{1}{N_1^\alpha} - \frac{\delta}{A} \right)^{-\frac{1}{\alpha}} .$$

In the settings of our experiments we have $N_1 \in \{124M, 210M\}$ and $D_1 \in \{10.24B, 20.48B\}$[5]. Plugging in the constants by Besiroglu et al. (2024)[6] and using $\delta = 0.01$, yields[7]

- Case 1: Fix $N_1 = N_2$. In this case, the scaling law results in $D_2 \in \{10.88B, 22.16B\}$ for $D_1 \in \{10.24B, 20.48B\}$, respectively. This means that we would need to train the models for $640M$ or $1.68B$ more tokens to match the adapted schedule.

- Case 2: Fix $D_1 = D_2$. In this case, we obtain $N_2 \in \{129.0M, 220.1M\}$ for $N_1 \in \{124M, 210M\}$. In other words, increasing the number of parameters by $5M$ or $10M$ would approximately result in the same loss after fixing the amount of tokens.

For both cases, the estimates from the scaling law match our general intuition and would require either noticeably training longer by $6 - 8\%$ or growing the model by $4 - 5\%$, in line with the argument at the end of Section 5.1. Also note that the (relative) additional cost implied by the Chinchilla law to obtain $0.01$ loss improvement grows with the (extended) training length $D_1$.

## B.6. Miscellaneous Plots

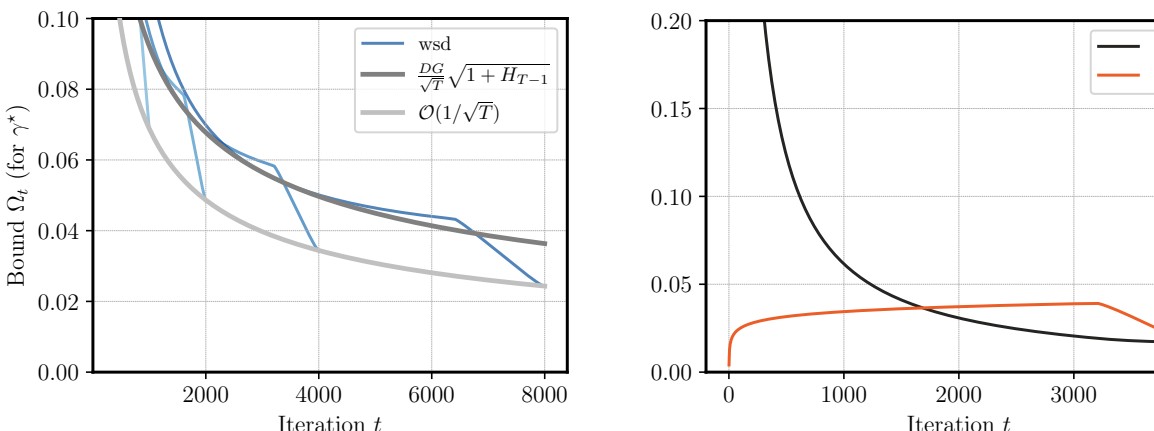

*Figure 20.* **(Left)** The benefit of cooldown is reflected in the absence of logarithmic terms. Dark grey marks the bound of the constant schedule. **(Right)** Plotting the individual terms of the bound $\Omega_t = \mathcal{T}_1/\gamma + \gamma \mathcal{T}_2$ with $\gamma = \gamma^\star$ for the `wsd` schedule. The sudden drop of the bound comes from the term $\gamma \mathcal{T}_2$.

---

[5]Batch size 50, two accumulation steps, two GPUs, sequence length 512, 100/200k steps.

[6]$A = 482.01, B = 2085.43, E = 1.8172, \alpha = 0.3478, \beta = 0.3658$.

[7]Note that the Chinchilla scaling laws were obtained in a different setup. In particular, we do not have access to the exact same dataset and tokenizer, which makes the scaling law not directly transferrable. However, our experiments are comparable in the general dataset composition (webcrawl data extended with other sources) and training task (decoder-only language models). Moreover, with the difference in vocabulary size (32k vs. 50k), we can scale the loss with the rough approximation of $\ln(32 \cdot 10^3)/\ln(50 \cdot 10^3) \approx 0.959$ to align the cross-entropy losses. This does not substantially change the results of this analysis.

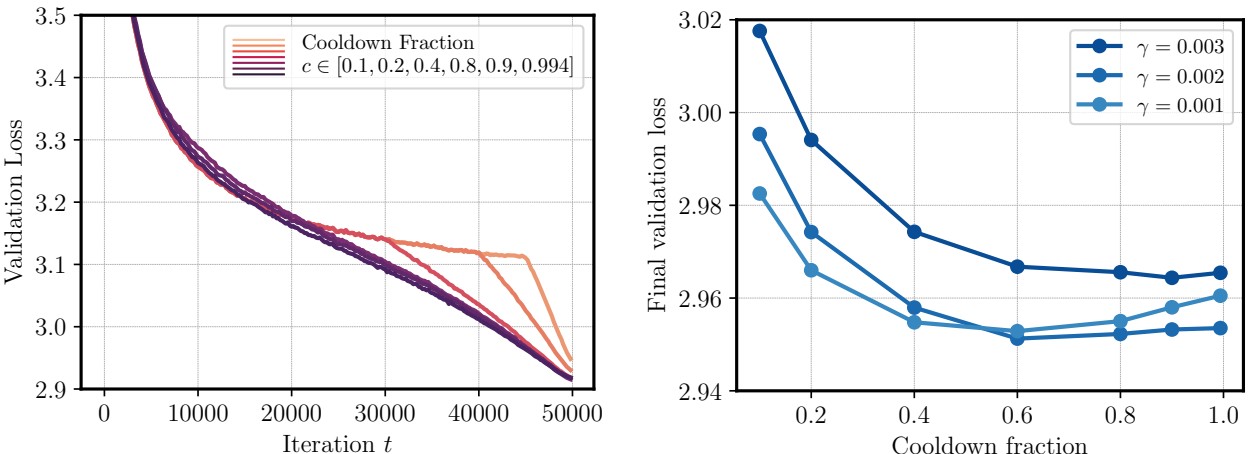

*Figure 21.* (**Left**) Analogous to Fig. 5 (right) with real training curves. We remove cooldown fraction 0.6 as its loss curve shows a spike and recovers only late. (**Right**) Analogous of Fig. 4 (right) with real training data that shows a parabola shape for fixed learning-rates.

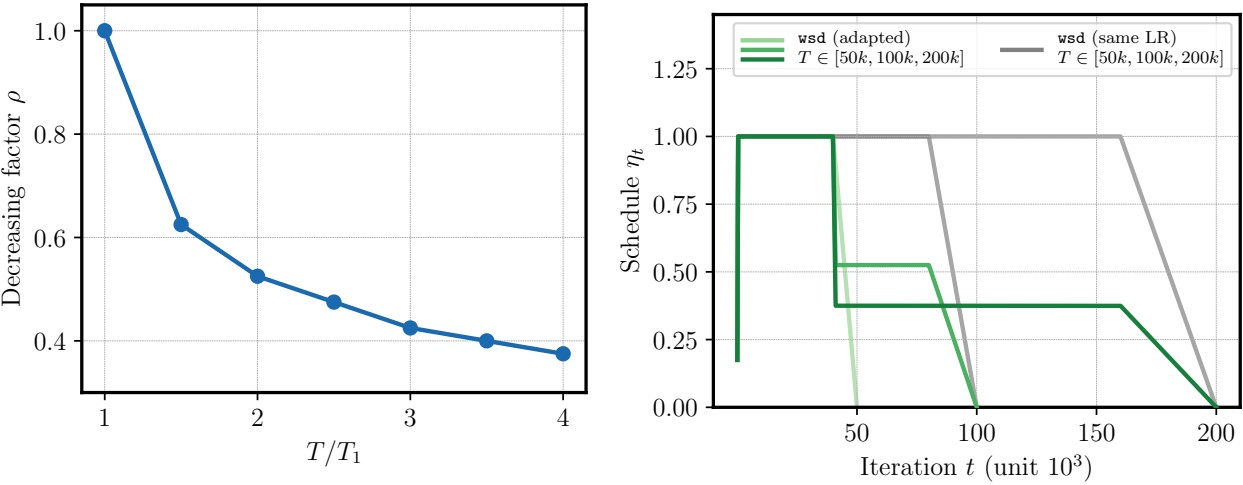

*Figure 22.* (**Left**) Decreasing factor computed with Corollary 3.3 for extended schedule up to $T$ (where $T_1$ is length of the short run). See Section 5.1, (B2) for details. (**Right**): Extended schedule for the training runs in Fig. 10; note that this schedule is multiplied by $\gamma = 0.001$.

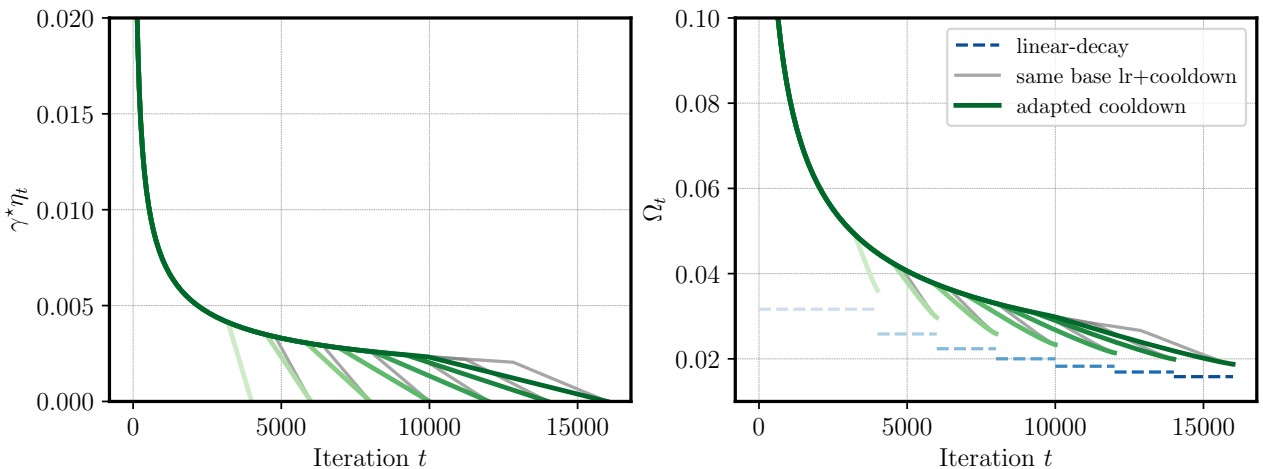

*Figure 23.* **(Left)** Transferring the `1/sqrt` schedule with linear cooldown from horizon $T_1 = 4000$ to $T_2 \in [1.5T_1, 4T_1]$. **(Right)** Adapting the cooldown length has only small benefits. Dashed horizontal lines mark bound for linear-decay schedule with tuned $\gamma^\star$. See Section 5.1, (B1) for details.

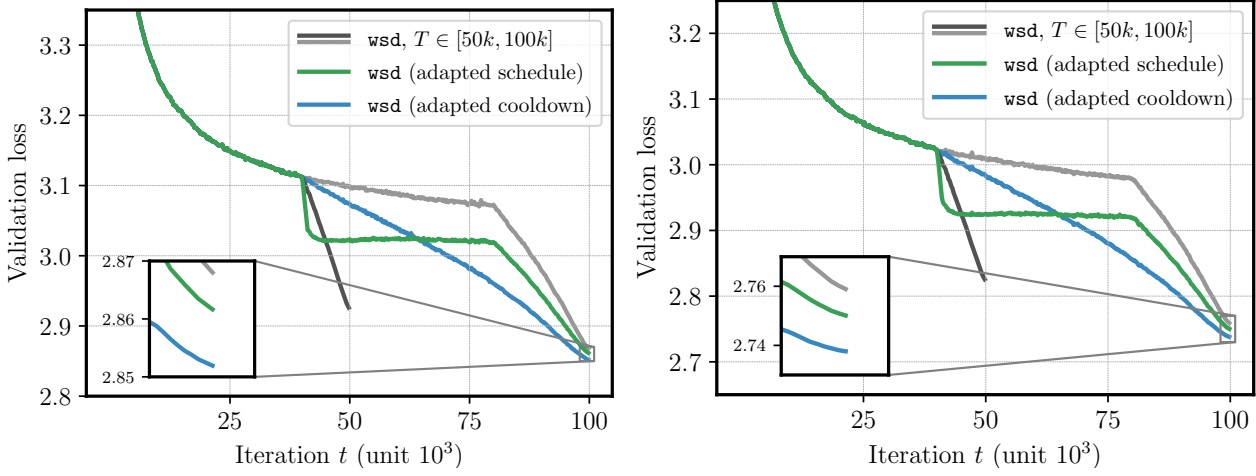

*Figure 24.* Experiment with adapted cooldown length for 124M model **(left)** and 210M **(right)**. See Section 5.1, (B1) for details.

# C. Additional Experiments

## C.1. Additional Experiments on `Imagenet`

In order to corroborate our findings on additional data domains and architectures, we conduct additional experiments on training `ResNet50` (He et al., 2016) on `Imagenet`. We train all models with `SGD` with heavy-ball momentum.

Training is done using the `timm` library (Wightman, 2019). All runs are using weight decay of $0.0001$, momentum $0.9$, batch size $4 \times 256$, and standard data augmentation techniques.[8]

Figs. 25 and 26 confirm our previous findings on this additional training task. **Note that** the `Imagenet` training is in a different regime as we are not training with a single pass. Hence, we expect the validation set metrics to be confounded by generalization effects beyond training loss only.

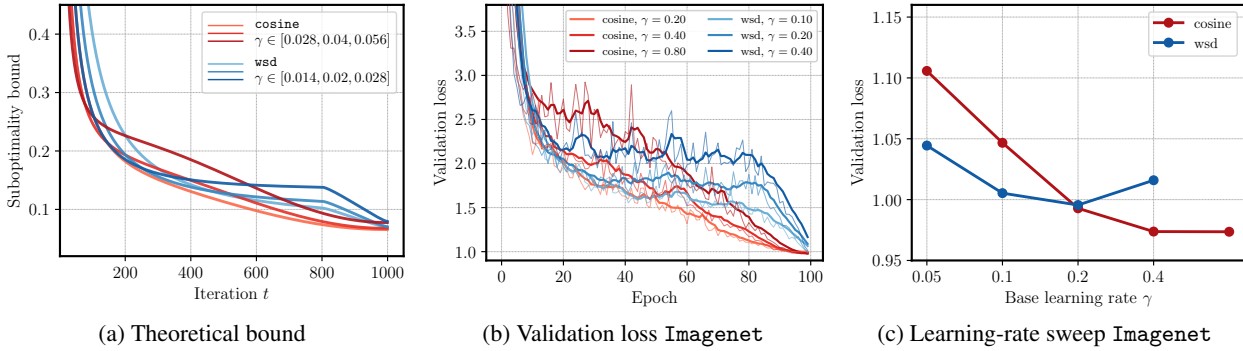

(a) Theoretical bound  (b) Validation loss `Imagenet`  (c) Learning-rate sweep `Imagenet`

*Figure 25.* The theoretical bound **(left)** for a range of base learning rates $\gamma$ (and setting $D = G = 1$) qualitatively matches the empirical (validation) loss curves for training `ResNet50` on `Imagenet` with `SGD` **(middle)**. (We display a running average over five epoch in thick to smoothen the plot, and the original data in thin.) **(Right)** We again find for the optimal base learning rate that it holds $\gamma^{\star}(\texttt{cosine}) \approx 2\gamma^{\star}(\texttt{wsd})$, as is predicted by the theory.

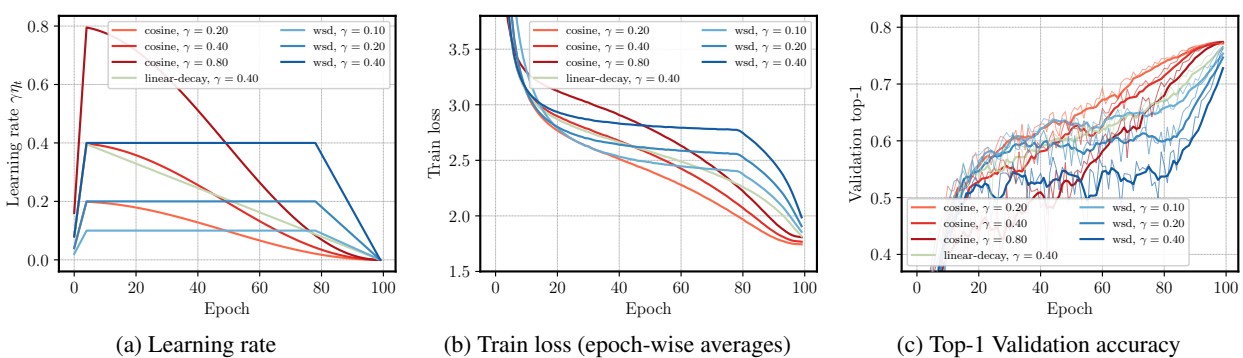

(a) Learning rate  (b) Train loss (epoch-wise averages)  (c) Top-1 Validation accuracy

*Figure 26.* Training `ResNet50` on `Imagenet` with `SGD`: Plotting the three best base learning rates $\gamma$ for both `cosine` and `wsd` schedule, as well as linear-decay schedule with learning rate transfer following Section 5.2. Sudden drop of training loss **(middle)** and increase of validation set accuracy **(right)** is clearly visible for `wsd` schedule. Linear-decay with zero-shot transfer of $\gamma^{\star}$ (multiplier $\exp(0.7) \approx 2$) improves over `wsd`, which confirms the findings of Section 5.2. For validation set metrics, we display a running average over five epoch in thick to smoothen the plot, and the original data in thin. Note that the final train loss of `wsd` appears slightly higher as we display the *epoch-wise average* of mini-batch losses; due to the steep descent of the loss at the end of training this slightly distorts the plot.

## C.2. Additional Experiments on `OpenWebText2`

We also train language models on a different dataset, namely `OpenWebText2`. Across three different model sizes we find (again) that the performance of `wsd` (with $20\%$ cooldown) matches the one of `cosine` when using a base learning rate

---

[8]In `timm` we set the configuration `hflip`$= 0.5$, `vflip`$= 0$, `crop-pct`$= 1.0$.

half as big (see Fig. 27). For `wsd`, the sudden drop in the loss is clearly visible across all model sizes and training lengths. Training details are identical to the `OpenWebText2` experiments of Hägele et al. (2024).

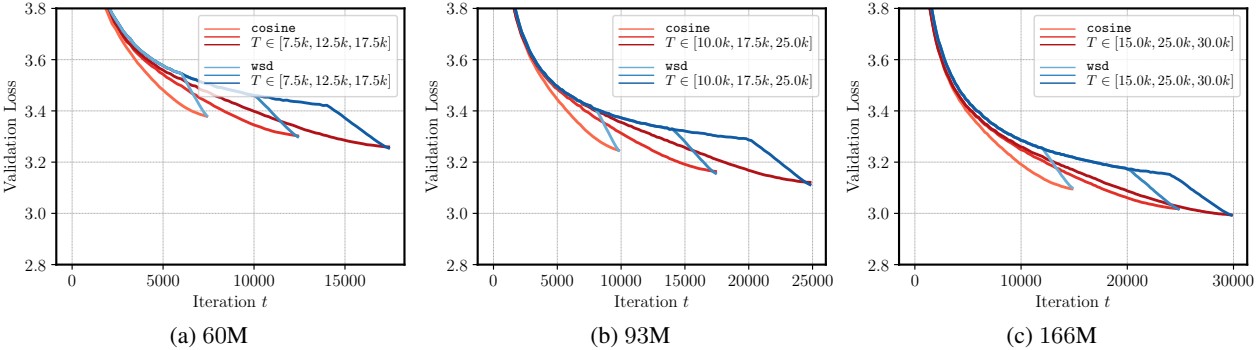

| (a) 60M | (b) 93M | (c) 166M |

*Figure 27.* Training three different model sizes on `OpenWebText2`. We observe the same characteristic drop of the loss for `wsd`, as well as matching performance of `wsd` and `cosine`. In each run the base learning rate of `cosine` is twice as large as for `wsd`.

### C.3. Computing the Bound-minimizing Schedule

In this section, we showcase how the schedule that minimizes the bound $\Omega_T$ in Theorem 3.1 can be computed. However, we stress that this has more of a purpose of illustration than practical applicability: in general, the gradient norms will depend on the schedule, and thus create an interdependence of the schedule and gradient norms (see also Defazio et al. (2023)).

However, we will show that – with constant gradient norm bounds ($G_t = 1$ for all $t \in \mathbb{N}$) – computing the schedule that minimizes $\Omega_T$ converges to the linear-decay schedule. Consider $\Omega_T(\eta_{1:T})$, the right-hand side in Theorem E.2, as a function of the schedule $\eta_{1:T} = (\eta_1, \ldots, \eta_T)$. We are interested in computing

$$\arg\min_{\eta_{1:T}} \Omega_T(\eta_{1:T}) \quad \text{subject to } \eta_1, \ldots, \eta_T > 0.$$

As we can not compute the above analytically, we resort to using projected gradient descent (with momentum) in order to approximate the solution. We implement $\Omega_T(\eta_{1:T})$ in `Pytorch` which allows us to use automatic differentiation. As starting point, we use a constant schedule $\eta_t = 1$ for all $t = 1, \ldots, T$. Fig. 28 (left) shows the value of the bound $\Omega_T$ over the optimization trajectory. We observe convergence of the optimized schedule to a linear-decay schedule (Fig. 28, right). Note that the shape of the schedule *as well as* its scale are optimized simultaneously; we plot the optimal bound for a linear-decay schedule and see that its value matches the bound at convergence. Though we did not observe any instabilities in the optimization procedure, at this point we can not verify whether the final point is a global minimum.

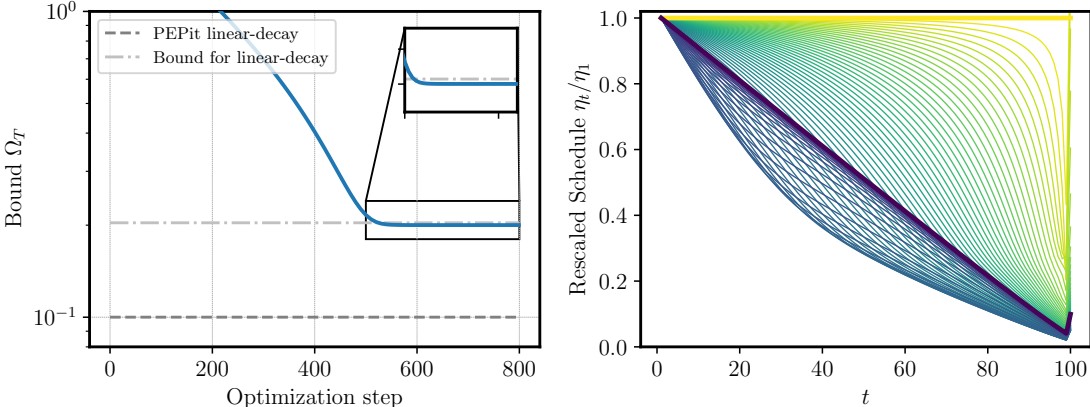

*Figure 28.* Optimizing the bound $\Omega_T$ with respect to the schedule $\eta_1, \ldots, \eta_T$. **(Left)** Convergence of the value of the bound, matching the bound of a linear-decay schedule. **(Right)** Optimization trajectory from constant schedule (yellow) to linear-decay (purple).

# D. Auxiliary Lemmas

**Lemma D.1** (Lemma 5 from Defazio et al. (2023)). *Let $(q_t)$ be any sequence, and let $(w_t)$ be a positive sequence. Then, for any $T \in \mathbb{N}$ it holds*

$$q_T = \frac{1}{\sum_{t=1}^{T} w_t} \sum_{t=1}^{T} w_t q_t + \sum_{k=1}^{T-1} \frac{w_k}{\sum_{t=k+1}^{T} w_t} \Big( \frac{1}{\sum_{t=k}^{T} w_t} \sum_{t=k}^{T} w_t (q_t - q_k) \Big).$$

**Lemma D.2.** *Let $l, T \in \mathbb{N}$ and $l \leq T$. It holds*

$$\sum_{s=1}^{T+1-l} s = \tfrac{1}{2}(T+2-l)(T+1-l), \quad \sum_{s=1}^{T+1-l} s^2 = \tfrac{1}{6}(2T+3-2l)(T+2-l)(T+1-l).$$

*Proof.* We refer to WolframAlpha: [link to first result], [link to second result]. □

**Lemma D.3.** *Let $t \in \mathbb{N}$. It holds*

$$\ln(t) \leq \ln(t+1) \leq \int_0^t \frac{1}{s+1} ds \leq \sum_{s=1}^{t} \frac{1}{s} = H_t \leq 1 + \ln(t).$$

# E. Missing Proofs

The following lemma will be the basic inequality for subsequently proving Theorem E.2; it is a standard result in the online learning and convex optimization literature (Zinkevich, 2003).

**Lemma E.1.** *Let $1 \leq k \leq T$ and let $u \in \mathbb{R}^d$ be measurable with respect to $x_k$. It holds*

$$\sum_{t=k}^{T} \eta_t \mathbb{E}[f(x_t) - f(u)] \leq \frac{1}{2} \mathbb{E}\|x_k - u\|^2 + \frac{1}{2} \sum_{t=k}^{T} \eta_t^2 \mathbb{E}\|g_t\|^2. \tag{13}$$

*Proof.* From the update rule (2) and property (4) we obtain

$$\|x_{t+1} - u\|^2 = \|x_t - u\|^2 - 2\eta_t \langle g_t, x_t - u \rangle + \eta_t^2 \|g_t\|^2$$
$$\leq \|x_t - u\|^2 - 2\eta_t \big[ f(x_t, s_t) - f(u, s_t) \big] + \eta_t^2 \|g_t\|^2.$$

Apply conditional expectation (conditioned on $t$) to obtain

$$\mathbb{E}\|x_{t+1} - u\|^2 \leq \|x_t - u\|^2 - 2\eta_t \big[ f(x_t) - f(u) \big] + \eta_t^2 \mathbb{E}\|g_t\|^2.$$

Apply total expectation (with respect to $t = 1, \dots, T$) and rearrange to obtain

$$\eta_t \mathbb{E}[f(x_t) - f(u)] \leq \tfrac{1}{2} \mathbb{E}\|x_t - u\|^2 - \tfrac{1}{2} \mathbb{E}\|x_{t+1} - u\|^2 + \frac{\eta_t^2}{2} \mathbb{E}\|g_t\|^2.$$

Sum from $t = k, \dots, T$ to obtain the final result:

$$\sum_{t=k}^{T} \eta_t \mathbb{E}[f(x_t) - f(u)] \leq \frac{1}{2} \mathbb{E}\|x_k - u\|^2 + \frac{1}{2} \sum_{t=k}^{T} \eta_t^2 \mathbb{E}\|g_t\|^2.$$

□

**Theorem E.2** (Thm. 10 from Defazio et al. (2023)). *Let the iterates $(x_t)$ be given by (2) with $\eta_t > 0$ for $t = 1, \dots, T$. Let $x_\star \in \mathbb{R}^d$ and $D := \|x_1 - x_\star\|$. Then, it holds*

$$\mathbb{E}[f(x_T) - f(x_\star)] \leq \frac{D^2}{2 \sum_{t=1}^{T} \eta_t} + \frac{\sum_{t=1}^{T} \eta_t^2 \mathbb{E}\|g_t\|^2}{2 \sum_{t=1}^{T} \eta_t}$$

$$+ \frac{1}{2} \sum_{k=1}^{T-1} \frac{\eta_k}{\sum_{t=k+1}^{T} \eta_t} \Big( \frac{1}{\sum_{t=k}^{T} \eta_t} \sum_{t=k}^{T} \eta_t^2 \mathbb{E}\|g_t\|^2 \Big).$$

*Remark* E.3.

(i) Note that the result of Theorem E.2 is an *anytime* result, in the sense that we can evaluate the right-hand side at any $T$ without the knowledge of $\eta_t$ for $t > T$.

(ii) A technical artifact of Theorem E.2 is that $x_T$ does not depend on $\eta_T$ by definition, however $\eta_T$ appears in the bound on the right-hand side. This is standard in the analysis of subgradient methods: in the proof, we bound $-\mathbb{E}\|x_{T+1} - x_\star\|^2 \leq 0$. If one carries through this term to the end, then we obtain multiple terms in the bound that depend on $\eta_T$.

Theorem 3.1 follows from applying Theorem E.2 with $\hat{\eta}_t := \gamma\eta_t$. We finally prove Theorem E.2.

*Proof.* First, apply Lemma E.1 with $u \to x_\star$ and $k \to 1$ to obtain

$$\sum_{t=1}^{T} \eta_t \mathbb{E}[f(x_t) - f(x_\star)] \leq \frac{1}{2}D^2 + \frac{1}{2}\sum_{t=1}^{T} \eta_t^2 \mathbb{E}\|g_t\|^2. \tag{14}$$

Define $q_t := \mathbb{E}[f(x_t) - f(x_\star)]$. Dividing by $\sum_{t=1}^{T} \eta_t$ gives

$$\frac{1}{\sum_{t=1}^{T} \eta_t} \sum_{t=1}^{T} \eta_t q_t \leq \frac{D^2}{2\sum_{t=1}^{T} \eta_t} + \frac{\sum_{t=1}^{T} \eta_t^2 \mathbb{E}\|g_t\|^2}{2\sum_{t=1}^{T} \eta_t}. \tag{15}$$

In order to apply Lemma D.1 with $w_t \to \eta_t$ we need to bound the term

$$\sum_{t=k}^{T} \eta_t (q_t - q_k) = \sum_{t=k}^{T} \eta_t \mathbb{E}[f(x_t) - f(x_k)]. \tag{16}$$

Thus, apply Lemma E.1 with $u \to x_k$ to obtain

$$\frac{1}{\sum_{t=k}^{T} \eta_t} \sum_{t=k}^{T} \eta_t [q_t - q_k] \leq \frac{1}{2\sum_{t=k}^{T} \eta_t} \sum_{t=k}^{T} \eta_t^2 \mathbb{E}\|g_t\|^2.$$

Now, combine Lemma D.1 with (15) and (16) to get

$$\mathbb{E}[f(x_T) - f(x_\star)] = q_T \leq \frac{D^2}{2\sum_{t=1}^{T} \eta_t} + \frac{\sum_{t=1}^{T} \eta_t^2 \mathbb{E}\|g_t\|^2}{2\sum_{t=1}^{T} \eta_t}$$
$$+ \frac{1}{2}\sum_{k=1}^{T-1} \frac{\eta_k}{\sum_{t=k+1}^{T} \eta_t} \left(\frac{1}{\sum_{t=k}^{T} \eta_t} \sum_{t=k}^{T} \eta_t^2 \mathbb{E}\|g_t\|^2\right).$$

$\square$

# F. Mirror Descent Analysis

In this section, we extend the bound from Theorem E.2 to the stochastic mirror-descent method.

**Notation.** *In this section only*, we denote with $\|\cdot\|$ an arbitrary norm (in contrast to the rest of the paper, where it denotes the standard Euclidean norm), and let $\|\cdot\|_*$ denote its dual norm, defined by $\|x\|_* := \sup_{z \in \mathbb{R}^d : \|z\| \leq 1} \langle x, z \rangle$.

Let $\psi : \mathbb{R}^d \to \mathbb{R}$ be a continuously differentiable function that is $\mu$-strongly convex with respect to $\|\cdot\|$. Define the Bregman divergence as

$$B_\psi(x, y) := \psi(x) - \psi(y) - \langle x - y, \nabla\psi(y) \rangle.$$

It follows $B_\psi(x, y) \geq \frac{\mu}{2}\|x - y\|^2$ from strong convexity of $\psi$. Further, we will need the following three-point-identity (Beck & Teboulle, 2003, Lem. 4.1): for any $x, y, z \in \mathbb{R}^d$ it holds

$$B_\psi(z, x) + B_\psi(x, y) - B_\psi(z, y) = \langle \nabla\psi(y) - \nabla\psi(x), z - x \rangle. \tag{17}$$

Now, the iterates of (stochastic) mirror descent are given by: for $\eta_t > 0$ and $g_t \in \partial f(x_t, s_t)$, compute

$$x_{t+1} = \arg\min_{y \in \mathbb{R}^d} \eta_t \langle g_t, y - x_t \rangle + B_\psi(y, x_t). \tag{18}$$

We will now prove a mirror descent version of Theorem E.2 (in fact, Theorem E.2 is a special case with $B_\psi(x, y) = \frac{1}{2}\|x - y\|^2$). To do so, we first follow standard steps in mirror-descent analysis (Beck & Teboulle, 2003) to obtain the basic inequality in Lemma F.1. In contrast to the classical mirror-descent analysis, we use this to prove a *last-iterate* bound in Theorem F.2.

**Lemma F.1.** *Let the iterates $(x_t)$ be generated by (18). Let $1 \leq k \leq T$ and let $u \in \mathbb{R}^d$ be measurable with respect to $x_k$. It holds*

$$\sum_{t=k}^{T} \eta_t \mathbb{E}[f(x_t) - f(u)] \leq \mathbb{E}[B_\psi(u, x_k)] + \frac{1}{2\mu} \sum_{t=k}^{T} \eta_t^2 \mathbb{E}\|g_t\|_*^2. \tag{19}$$

*Proof.* For fixed $y$, we have $\nabla_x B_\psi(x, y) = \nabla\psi(x) - \nabla\psi(y)$. Thus, optimality conditions of (18) are

$$0 = \eta_t g_t + \nabla\psi(x_{t+1}) - \nabla\psi(x_t). \tag{20}$$

Then, we have

$$\eta_t[f(x_t, s_t) - f(u, s_t)] \leq \eta_t \langle x_t - u, g_t \rangle$$
$$= \underbrace{\langle u - x_{t+1}, \nabla\psi(x_t) - \nabla\psi(x_{t+1}) - \eta_t g_t \rangle}_{:=s_1} + \underbrace{\langle u - x_{t+1}, \nabla\psi(x_{t+1}) - \nabla\psi(x_t) \rangle}_{:=s_2} +$$
$$\underbrace{\langle x_t - x_{t+1}, \eta_t g_t \rangle}_{:=s_3}.$$

From (20), we have $s_1 = 0$. From (17), we have $s_2 = B_\psi(u, x_t) - B_\psi(u, x_{t+1}) - B_\psi(x_{t+1}, x_t)$. From the (generalized) Cauchy-Schwarz inequality combined with Young's inequality, we have

$$s_3 \leq \frac{\mu}{2}\|x_{t+1} - x_t\|^2 + \frac{\eta_t^2}{2\mu}\|g_t\|_*^2.$$

Using that $-B_\psi(x_{t+1}, x_t) \leq -\frac{\mu}{2}\|x_{t+1} - x_t\|^2$, we obtain

$$\eta_t[f(x_t, s_t) - f(u, s_t)] \leq B_\psi(u, x_t) - B_\psi(u, x_{t+1}) + \frac{\eta_t^2}{2\mu}\|g_t\|_*^2.$$

Taking conditional expectation, we have $\mathbb{E}[f(x_t, s_t) - f(u, s_t)] = f(x_t) - f(u)$. Finally, rearrange, take total expectation and sum from $t = k, \ldots, T$. Using that $B_\psi(u, x_{T+1}) \geq 0$, we obtain

$$\sum_{t=k}^T \eta_t \mathbb{E}[f(x_t) - f(u)] \leq \mathbb{E}[B_\psi(u, x_k)] + \frac{1}{2\mu} \sum_{t=k}^T \eta_t^2 \mathbb{E}\|g_t\|_*^2.$$

$\square$

Now, repeating the proof of Theorem E.2, but applying Lemma F.1 instead of Lemma E.1, we obtain the following bound.

**Theorem F.2.** *Let the iterates* $(x_t)$ *be given by stochastic mirror descent* (18) *with* $\eta_t > 0$ *for* $t = 1, \ldots, T$. *Let* $x_\star \in \mathbb{R}^d$. *Then, it holds*

$$\mathbb{E}[f(x_T) - f(x_\star)] = \frac{\mathbb{E}[B_\psi(x_\star, x_1)]}{\sum_{t=1}^T \eta_t} + \frac{\sum_{t=1}^T \eta_t^2 \mathbb{E}\|g_t\|_*^2}{2\mu \sum_{t=1}^T \eta_t}$$
$$+ \frac{1}{2\mu} \sum_{k=1}^{T-1} \frac{\eta_k}{\sum_{t=k+1}^T \eta_t} \left( \frac{1}{\sum_{t=k}^T \eta_t} \sum_{t=k}^T \eta_t^2 \mathbb{E}\|g_t\|_*^2 \right).$$

# G. Analysis of `wsd` schedule

We first state Theorem 3.4 in its full version.

**Theorem G.1.** *Let $1 \leq T_0 < T$. Assume that $\eta_t = 1$ for $t < T_0$ and $\eta_t = 1 - \frac{t - T_0}{T + 1 - T_0}$ for $T \geq t \geq T_0$. Further, assume that (A3) holds with $G_t = G$ for some $G > 0$ for all $t \in \mathbb{N}$. Then, for $\gamma = \gamma^\star$ from Corollary 3.3, we have*

$$\mathbb{E}[f(x_T) - f(x_\star)] \leq$$

$$DG\sqrt{\frac{4}{T + T_0}\left[\frac{2}{3} + \frac{T + 2T_0}{3(T + T_0)} + H_{T + T_0 - 2} - H_{T - T_0 + 1} - \frac{(T - T_0)(T_0 - 1)}{3(T - T_0 + 2)(T + T_0)} + \frac{1}{(T - T_0)^2} + \frac{H_{T - T_0 - 1}}{T - T_0 + 1}\right]}.$$

Note that for large $T$, we have $H_{T - T_0 - 1} = \mathcal{O}(\ln(T - T_0 - 1)) = o(T - T_0 + 1)$. Thus, the last two terms can be summarized with $o(1)$ as $T \to \infty$.

In this section, we give further interpretation the bound in Theorem G.1 in comparison to the constant and linear-decay schedules.

In Section 3.2, we derived that for large $T$ and $T_0 = \beta T$, the bound for `wsd` is approximately

$$\mathbb{E}[f(x_T) - f(x_\star)] \lesssim \frac{DG}{\sqrt{T}} \cdot \sqrt{\frac{4}{1 + \beta}\left[\frac{2}{3} + \frac{1 + 2\beta}{3(1 + \beta)} - \frac{\beta}{3(1 + \beta)} + H_{(1 + \beta)T - 2} - H_{(1 - \beta)T + 1}\right]}.$$

To obtain concrete numbers, plugging in $\beta = 0.8$ (that is, 20% cooldown) for `wsd`, and obtain

$$\mathbb{E}[f(x_T) - f(x_\star)] \lesssim \frac{DG}{\sqrt{T}} \cdot \sqrt{0.9 + 2.2(H_{1.8T - 2} - H_{0.2T + 1})}.$$

For example, if $T = 10^5$, then $2.2(H_{1.8T - 2} - H_{0.2T + 1}) \approx 4.39$. In comparison, we have:

- constant schedule: the bound is $\frac{DG}{\sqrt{T}} \cdot \sqrt{1 + H_{T - 1}}$.

- linear-decay schedule: the same bound from Corollary 3.3 results in $(2 + \frac{H_{T - 1} - 2/3}{T + 1})\frac{DG}{\sqrt{T}}$ (Defazio et al., 2023, Thm. 13). However, with a different (but less general) proof technique one can show the tighter bound $\frac{DG}{\sqrt{T}}$ (Defazio et al., 2023, Cor. 2), which is actually worst-case optimal for convex, Lipschitz problems (Zamani & Glineur, 2023).

Again for $T = 10^5$, we have $H_{T - 1} \approx 12.09$ and $(2 + \frac{H_{T - 1} - 2/3}{T + 1}) \approx 2.0001$. In conclusion, for this specific $T$ the constant of the bound is roughly twice for `wsd` compared to linear-decay, and $1/3$ compare to a constant schedule.

Finally, we prove Theorem G.1.

*Proof of Theorem G.1.* From Corollary 3.3, we have

$$\mathbb{E}[f(x_T) - f(x_\star)] \leq 2\sqrt{\mathcal{T}_1(\eta_{1:T}, D, T)\mathcal{T}_2(\eta_{1:T}, G_{1:T}, T)}.$$

Thus, the rest of the proof will compute an upper bound of the right-hand side. First, for $1 \leq l \leq T$, we compute

$$l \geq T_0 : \quad \sum_{t=l}^{T} \eta_t = \frac{1}{T + 1 - T_0}\sum_{t=l}^{T}(T + 1 - t) = \frac{1}{T + 1 - T_0}\sum_{s=1}^{T + 1 - l} s = \frac{(T + 2 - l)(T + 1 - l)}{2(T + 1 - T_0)}.$$

Here we made the change of variable $T + 1 - t \to s$ and used Lemma D.2. Similarly, we get

$$l < T_0 : \quad \sum_{t=l}^{T} \eta_t = \sum_{t=l}^{T_0 - 1} \eta_t + \sum_{t=T_0}^{T} \eta_t = T_0 - l + \sum_{t=T_0}^{T} \eta_t = T_0 - l + \frac{(T + 2 - T_0)(T + 1 - T_0)}{2(T + 1 - T_0)}$$
$$= \tfrac{1}{2}[T + 2 + T_0 - 2l].$$

Note that this expression is still correct if we would plug in $l = T_0$. Next, we compute the sum of squares. We start again with

$$l \geq T_0 : \quad \sum_{t=l}^{T} \eta_t^2 = \frac{1}{(T+1-T_0)^2} \sum_{s=1}^{T+1-l} s^2 = \frac{(2T+3-2l)(T+2-l)(T+1-l)}{6(T+1-T_0)^2}.$$

And similarly

$$l < T_0 : \quad \sum_{t=l}^{T} \eta_t^2 = \sum_{t=l}^{T_0-1} \eta_t^2 + \sum_{t=T_0}^{T} \eta_t^2 = T_0 - l + \frac{(2T+3-2T_0)(T+2-T_0)(T+1-T_0)}{6(T+1-T_0)^2}$$

$$= T_0 - l + \frac{(2T+3-2T_0)(T+2-T_0)}{6(T+1-T_0)}$$

$$= T_0 - l - \frac{2T+5-2T_0}{6} + \frac{1}{6(T+1-T_0)}$$

$$= \tfrac{1}{6}\Big[2T + 4T_0 + 5 - 6l + \frac{1}{(T+1-T_0)}\Big].$$

Here, we used that

$$(2T+3-2T_0)(T+2-T_0) = (2T+5-2T_0)(T+2-T_0) - 2(T+2-T_0)$$
$$= (2T+5-2T_0)(T+1-T_0) + (2T+5-2T_0) - 2(T+2-T_0)$$
$$= (2T+5-2T_0)(T+1-T_0) + 1.$$

Again, the expression we obtain for $l < T_0$ is correct if we would plug in $l = T_0$. Now we can try to compute the bound. We start with the easy ones: as $G_t = G > 0$ for all $t \in \mathbb{N}$, we obtain

$$\mathcal{T}_1(\eta_{1:T}, D, T) = \frac{1}{2\sum_{t=1}^{T} \eta_t} D^2 = \frac{D^2}{T+T_0},$$

$$\frac{1}{2\sum_{t=1}^{T} \eta_t}\Big(\sum_{t=1}^{T} \eta_t^2 G_t^2\Big) = \frac{G^2}{2} \frac{\tfrac{1}{6}[2T + 4T_0 + 5 - 6] + \frac{1}{6(T+1-T_0)}}{\tfrac{1}{2}(T+T_0)} \tag{21}$$

$$= \frac{G^2}{6(T+T_0)}\Big[2T + 4T_0 - 1 + \frac{1}{T+1-T_0}\Big] \leq \frac{G^2(T+2T_0)}{3(T+T_0)}.$$

The last inequality is due to $T_0 < T$ and thus $1 > \frac{1}{T+1-T_0}$. Next, for $k = 1, \ldots, T-1$, we need to compute

$$\frac{\eta_k}{\sum_{t=k+1}^{T} \eta_t}\Big(\frac{1}{\sum_{t=k}^{T} \eta_t} \sum_{t=k}^{T} \eta_t^2 G_t^2\Big).$$

Again, as $G_t = G > 0$ for all $t \in \mathbb{N}$, we omit $G$ for now, and start with the case $k \geq T_0$:

$$\frac{\eta_k}{\sum_{t=k+1}^{T} \eta_t}\Big(\frac{1}{\sum_{t=k}^{T} \eta_t} \sum_{t=k}^{T} \eta_t^2\Big) = \frac{\frac{T+1-k}{T+1-T_0} \cdot \frac{(2T+3-2k)(T+2-k)(T+1-k)}{6(T+1-T_0)^2}}{\frac{(T+2-k)(T+1-k)^2(T-k)}{4(T+1-T_0)^2}} = \frac{2(2T+3-2k)}{3(T+1-T_0)(T-k)}.$$

Now, if $k < T_0$:

$$\frac{\eta_k}{\sum_{t=k+1}^{T} \eta_t}\Big(\frac{1}{\sum_{t=k}^{T} \eta_t} \sum_{t=k}^{T} \eta_t^2\Big) = \frac{1 \cdot \tfrac{1}{6}\big[2T + 4T_0 + 5 - 6k + \frac{1}{(T+1-T_0)}\big]}{\tfrac{1}{4}[T + T_0 - 2k][T + 2 + T_0 - 2k]}$$

$$= \frac{2\big[2T + 4T_0 + 5 - 6k + \frac{1}{(T+1-T_0)}\big]}{3[T + T_0 - 2k][T + 2 + T_0 - 2k]}.$$

Now, compute

$$\sum_{k=1}^{T-1} \frac{\eta_k}{\sum_{t=k+1}^{T} \eta_t} \Big( \frac{1}{\sum_{t=k}^{T} \eta_t} \sum_{t=k}^{T} \eta_t^2 \Big) =$$

$$\frac{2}{3} \Big[ \underbrace{\sum_{k=1}^{T_0-1} \frac{\big[2T + 4T_0 + 5 - 6k + \frac{1}{(T+1-T_0)}\big]}{[T + T_0 - 2k][T + 2 + T_0 - 2k]}}_{:=\Omega_1} + \underbrace{\sum_{k=T_0}^{T-1} \frac{(2T + 3 - 2k)}{(T + 1 - T_0)(T - k)}}_{:=\Omega_2} \Big] =: (*).$$

Then, it holds [link to proof]

$$\Omega_2 = \sum_{k=T_0}^{T-1} \frac{(2T + 3 - 2k)}{(T + 1 - T_0)(T - k)} = \frac{2T - 2T_0 + \frac{3}{T-T_0} + 3H_{T-T_0-1}}{T - T_0 + 1}.$$

To simplify this term a bit, we estimate

$$\frac{2}{3} \Omega_2 \le \frac{4}{3} + \frac{2}{(T - T_0)^2} + \frac{2H_{T-T_0-1}}{T - T_0 + 1}.$$

For $\Omega_1$, we can bound the nominator by

$$2T + 4T_0 + 5 - 6k + \frac{1}{T+1-T_0} = 3(T + 2 + T_0 - 2k) - (T - T_0 + 1) + \frac{1}{T+1-T_0}$$

$$\le 3(T + 2 + T_0 - 2k) - (T - T_0),$$

where for the second term we bound $\frac{1}{T+1-T_0} \le 1$ due to $T_0 \le T$. It holds [link to proof]

$$\sum_{k=1}^{T_0-1} \frac{1}{[T + T_0 - 2k][T + 2 + T_0 - 2k]} = \frac{T_0 - 1}{(T - T_0 + 2)(T + T_0)}.$$

Therefore, defining $\Omega_3 := \frac{(T-T_0)(T_0-1)}{(T-T_0+2)(T+T_0)}$ we get

$$\Omega_1 \le \Big( \sum_{k=1}^{T_0-1} \frac{3}{T + T_0 - 2k} \Big) - \Omega_3 = 3(H_{T+T_0-2} - H_{T-T_0+1}) - \Omega_3,$$

where we used $T \ge T_0$. Altogether, we have

$$(*) = \frac{2}{3}(\Omega_1 + \Omega_2) \le 2(H_{T+T_0-2} - H_{T-T_0+1}) - \frac{2}{3}\Omega_3 + \frac{4}{3} + \frac{2}{(T - T_0)^2} + \frac{2H_{T-T_0-1}}{T - T_0 + 1}.$$

Multiplying this term with $\frac{G^2}{2}$, we can plug the result in (21) to finally derive the bound

$$2\sqrt{\mathcal{T}_1(\eta_{1:T}, D, T)\mathcal{T}_2(\eta_{1:T}, G_{1:T}, T)} \le$$

$$2DG\sqrt{\frac{1}{T+T_0}\Big[\frac{T+2T_0}{3(T+T_0)} + H_{T+T_0-2} - H_{T-T_0+1} - \frac{(T-T_0)(T_0-1)}{3(T-T_0+2)(T+T_0)} + \frac{2}{3} + \frac{1}{(T-T_0)^2} + \frac{H_{T-T_0-1}}{T-T_0+1}\Big]}.$$

$\square$

