# OpenReview forum: "The Surprising Agreement Between Convex Optimization Theory and Learning-Rate Scheduling for Large Model Training"
_ICML.cc/2025/Conference — ICML 2025 poster_

### Official Review · Reviewer_iNQu · 2025-03-05

**Overall Recommendation:** 2

**Summary:**

This paper proposes to find optimal settings of warmup-stable-decay (wsd) learning rate (LR) scheduler for training large language models (LLMs) from a theoretical inspiration. The wsd scheduler is a piecewise function by training iterations, the first part is a constant function (named based LR) and the second part is a linear function that decays the LR with a ratio proportional to iteration. Thus, the authors aim to investigate the configuration of two controllable hyper-parameters, i.e., base LR and a fraction of the decay period to the entire training iterations, by minimizing the convergence rate of the wsd scheduler.

First, the convergence rate is derived from minimizing the last loss with gradient descent and the wsd scheduler. The optimal base LR and optimal convergence rate are derived by solving a convex optimization problem of convergence rate minimization with respect to base LR. Further, an iteration-free rate is derived with respect to the fraction. The authors utilize several simulation results to demonstrate the relevance of the proposed optimal rate to base LR and the fraction. The results show that the optimal base LR decreases with training iterations but increases with the fraction's value.

Then, the authors try to apply the observations of simulation results in ideal convex optimization to real-world LLM training. Two simple strategies, i.e., decrease base LR and increase fraction rate, are proposed. Simulation results demonstrate the effectiveness of the synthetic convex optimization problem. Experimental results on 0.1B and 0.2B Llama-style transformers show the improvement of the proposed method. Moreover, the authors propose to fit a convergence rate function from several running results with the wsd scheduler, where the function is defined from the inspiration of the theoretical optimal rate, to demonstrate the alignment of the proposed method to the real-world LLM training.

## update after rebuttal
I keep my rating due to the following main concerns:
This paper is within the scope of the study of scaling law. Thus, the only way to demonstrate its effectiveness is by experimental results on heterogeneous LLMs, especially for large LLMs. However, the selected LLMs are only 124/210M Llama-style transformers, and there are no additional experiments on large LLMs in rebuttal.

**Claims And Evidence:**

Yes.

**Essential References Not Discussed:**

No.

**Experimental Designs Or Analyses:**

The experimental designs are not enough to demonstrate the so-called effectiveness of LLMs, since the selected LLMs are extremely small, less than 1B.

**Methods And Evaluation Criteria:**

Not enough.
First, training loss is not an important criterion for LLMs. Thus, the optimality of loss in training LLMs is not a convincing motivation for the so-called requirement of optimal learning rate.
Second, the selected LLMs are too small. The selected LLMs are less than 1B, which is not sufficient to demonstrate the effectiveness of the proposed method for LLMs.

**Other Comments Or Suggestions:**

Typo: "The wsd schedule." paragraph, last sentence.

**Other Strengths And Weaknesses:**

Strength:
1. This paper conducts many simulations and experiments on two small LLMs.
2. This paper proposes a simulation-based learning rate (LR) scheduling method. By sampling real-world running results from different LR configurations, a function inspired by theoretical optimal LR is proposed to fit the real-world running results. Then, the optimizer to the function yields the "optimal" LR configuration.

Weakness:
The significance of this paper is limited. This paper tries to demonstrate that the empirical learning rate (LR) scheduling methods in training LLM align with its theoretical optimal solutions, which include three main components, i.e., theoretical proofs, LLM training, and alignment. This paper's soundness is poor in all three parts.
1. The theoretical proofs are based on a strict convex optimization scenario, which is far from the high dimensional non-convex optimization scenario in neural network (NN) training. The assumptions are too strict that the loss function is extremely smooth, in which the function's gradient can be bounded by constants (A3 in the paper). Thus, it may be the reason for the results in Fig 4, where a larger decay rate leads to under-training (large loss).
2. The selected LLMs are too small to demonstrate the effectiveness of the proposed method for LLMs. The simulation results are trivial under some specific loss functions and settings. First, most simulation results are based on extreme smooth loss ($G=1$) and a close initialization to optimum ($D=1$). Moreover, most observations (such as the specific settings of the decay fraction rate from Fig. 10, 11) are based on the 124M and 210M Llama-style transformers, which may not hold on large LLMs.
3. The alignment is problematic. The connection of the proposed theorem to neural network training is achieved by approximating a linear function of the hyper-parameters in the theorem, which is not convincing since there are no standard solutions for the function approximation for all NN models. Further, most simulations and experiments are designed to evaluate the proposed method for optimal base LR the wsd LR decay method. However, the most crucial part is whether such a theoretical upper bound of loss is close enough, which is not evaluated in the paper. For example, from Fig. 12 (b), most losses are above the approximated function, which violates the upper bound requirement of the function.

**Questions For Authors:**

1. Does $\gamma$ in Eq. (9) represent $\gamma^{\star}$?
2. What is the $f$ in all simulation results?
3. What's the meaning of Fig. 8？ How to distinguish the training before $T_1$ and after $T_1$? The illustrated result looks like the overrides of the former training processes.
4. In Fig. 17, from the left part, the cosine scheduler converges faster than the wsd scheduler. However, the observation seems to be the opposite of the right part.

**Relation To Broader Scientific Literature:**

This paper aims to improve an existing learning rate scheduling method from theoretical inspirations of ideal convex optimization. However, the gap between strict assumptions and real-world neural network training is not well-addressed.

**Theoretical Claims:**

Yes. The proofs are correct.

---

> ### Author Rebuttal · Authors · 2025-03-31
>
> We thank the reviewer for the feedback and suggestions, and address each question below in detail. We hope that after clarifying all concerns we can improve the reviewer's rating of our submission.
>
> * *First, training loss is not an important criterion for LLMs*: For all LLM experiments in our paper, **we already plot the validation set loss**. We also comment on the relationship between train and test/validation loss in the last paragraph of the Limitation section.
> * *the gap between strict assumptions and real-world neural network training is not well-addressed*: The limitations of these assumptions are explicitly addressed in Section 6. As the title suggests, one of the insights of this paper is that theory and practice match very well, even though the assumptions from theory are not satisfied.
> * *the loss function is extremely smooth*: Can you explain what is meant with the term 'extremely smooth'? In fact, our theoretical results hold for **non-smooth functions** with bounded gradients (e.g. the absolute value function would satisfy this, but it’s not smooth).  Thus we are confused by this comment. We also remark that we set $D=1$ and $G=1$ in the simulation purely for simplicity (see details further below). The bound holds for any other values. (In fact, for the LLM training setup, the gradient norms are below one except for the first ~hundred steps; we will provide a plot of the empirical gradient norms in the final version.)
> * *is achieved by approximating a linear function of the hyper-parameters in the theorem*: We did not not understand this comment. Could you explain which linearization is meant here?
> * *Fig 4, where a larger decay rate leads to under-training (large loss)*: we are confused about this comment, as Fig 4 shows that larger decays lead to lower losses if the learning rate is tuned. Could the reviewer expand on this, so that we can hopefully answer the question?
> * *The alignment is problematic*: We kindly disagree: our theoretical findings are verified on multiple schedules+model sizes+time horizons, and we show that **multiple aspects of the theory match the experiments** (e.g. schedule adaptation for longer runs, effect of cooldown length, LR transfer across cooldown lengths etc.).
> * *LLMs are less than 1B, which is not sufficient*: Our experiments require a high number of individual runs, for example to sweep the learning rate (e.g. *only* Figure 12b requires around 20 runs). With our computational budget it would not be feasible to execute this amount of runs on a >10x larger scale. Besides that, scaling law research has repeatedly shown that loss improvements transfer to larger scales highly predictably in this regime (see https://arxiv.org/pdf/2203.15556). We also point out that for the central message of our paper (“convex bounds match empirical behaviour in deep learning”) any scale would be sufficient to prove the point.
>
> On Questions:
> 1) In Equation (9), $\gamma$ is arbitrary, and not necessarily equal to $\gamma^\star$.
> 2) The simulations simply evaluate the bound $\Omega_t$ for different schedules and learning rates. For this it is not necessary to specify $f$, but only constants $D$ and $G$ which we set to 1 in the simulations (purely for simplicity). The choice of $D$ and $G$ only affects the scale of the bound and learning rates, but not its shape or scaling in $T$.
> 3) As the caption of Fig 8 explains, T_1=400 and we test several values of T_2. Indeed the goal is to reuse the training up to iteration T_1, as it is explained in the beginning of Section 5.1
> 4) The right plot is only the iterate path, and one can not exactly read off the timescale of it. However, for (i) cosine at the step where its path makes a sharp left-wards turn and (ii) for WSD before cooldown are roughly similar and this matches the contours in the right plot.

---

> > ### Comment · Reviewer_iNQu · 2025-04-02
> >
> > Clarification:
> > 1. "extremely smooth" refers to the Lipschitz continuous setting, i.e., $D=1$ and $G=1$ is an over-simplified setting, which will be almost inconstant with NN training.
> > 2. "linearization". As stated in the review comment, the main technique for the alignment between the ideal theorem and LLM training is approximating a linear function as a learning rate scheduler in Figure 12.
> >
> > Weakness:
> > 1. "validation set". In the rebuttal, the authors still don't explain the motivation of approximation a learning rate from training loss but not validation loss.
> > 2.  "alignment". The performance of small LLMs is not enough to demonstrate the effectiveness of such approximation-based method.

---

> > > ### Author Response · Authors · 2025-04-03
> > >
> > > Dear reviewer,
> > >
> > > thank you for clarifying your questions. We respond to each point below. If any concern remains, we kindly ask the reviewer to elaborate why our responses are not satisfying.
> > >
> > > * As stated in the rebuttal, the choice of D and G in the simulations is purely for simplicity, and the bounds holds for any other values. Two further remarks:
> > > 1) For the theoretical predictions on optimal LR, we made sure that the choice of D and G has no impact on the result (see line 373 right column). We invite the reviewer to verify this by running the scripts with other values for D and G (scripts for all analyses are contained in our supplementary material).
> > > 2) In fact, the assumption that gradient norms are bounded by one is not unrealistic in LLM training: for example, see the gradient norm logs for the Olmo2-7B model, where the gradient norm is always below one except for the first few iterations: see https://wandb.ai/ai2-llm/OLMo-2-1124-7B, and then search for "optim/total_grad_norm" in the panel search box.
> > >
> > > * There is no linearization in Figure 12. You might refer to the fact that we fit a function to the loss values (as function of learning rate) on the right plot. This procedure is in detail described in the appendix (see line 792 ff.); however, the function we fit is not linear, and well-motivated from theory.
> > >
> > > * "Validation set": the theoretical bound is the expected loss, which in general describes the training loss. We argue in the last paragraph of Section 6 that in single-pass training this coincides with the test loss (see the reference Aitchison, 2024). To the best of our knowledge, there are no convergence bounds that directly estimate the test loss. Thus, we do not comprehend how we could improve our submission in this regard.
> > >
> > > * "LLMs too small" Please read our response in the rebuttal. It seems arbitrary to us to draw the line for significance at 1B, and is concerning if papers are rejected solely based on this argument.
> > >
> > >
> > > References:
> > >
> > > Aitchison, 2024: Why you don't overfit, and don't need Bayes if you only train for one epoch. https://arxiv.org/abs/2411.14478

---

### Official Review · Reviewer_CJZu · 2025-03-07

**Overall Recommendation:** 4

**Summary:**

This paper studies learning-rate schedules in large model training, by bridging a new theoretical convergence analysis to empirical observations.

In particular, the first contribution (observation) is that for two popular schedules (cosine and wsd), the empirical loss curves of large model training, which is a non-convex optimization problem, has a similar shape with the theoretical bound predicted by the parameters in the convex case.

The second contribution is a new theoretical convergence analysis of wsd in the convex case, which roughly shows an $\log T$ improvement over the best constant learning rate, providing an explanation to the success of wsd.

The final contribution is an application of the theory, for continue training and learning rate transfer.

## update after rebuttal

I have read other reviews and rebuttals. Though I agree with Reviewer rpAe that the results presented are not significant enough in either theoretical or empirical side, the new link does seem interesting and promising. I will keep my score.

**Claims And Evidence:**

Yes.

**Essential References Not Discussed:**

As far as I know, no.

**Experimental Designs Or Analyses:**

Yes.

**Methods And Evaluation Criteria:**

Yes.

**Other Comments Or Suggestions:**

NAN

**Other Strengths And Weaknesses:**

The paper is well-written.

The theoretical result might not be a deep one from a technical perspective. However, it's very important to provide a rigorous explanation for a popular learning rate scheduler.

**Questions For Authors:**

Is it possible to perform a rigorous convergence rate analysis for cosine?

**Relation To Broader Scientific Literature:**

I think the results of the paper might be interesting to only only empirical ML researchers, but also optimization researchers.

**Theoretical Claims:**

Yes. The only new theoretical result is Theorem 3.4.

---

> ### Author Rebuttal · Authors · 2025-03-28
>
> We thank the reviewer for acknowledging the theoretical and empirical contributions of our paper. We are delighted that the reviewer considers the paper to be interesting to the optimization and ML community. Regarding the questions:
>
> * Convergence rate of cosine: unfortunately, due to the form of the cosine schedule it is quite difficult to explicitly derive the bound. We tried several simplifications and have not reached a nice result yet. However, from the simulations we conduct, the bound of cosine is almost the same as the one for linear-decay, for which we can explicitly derive the bound (see lines 1240 ff.).

---

### Official Review · Reviewer_hqnc · 2025-03-09

**Overall Recommendation:** 4

**Summary:**

This paper establishes a novel connection between empirical learning-rate schedules (e.g., cosine, WSD) used in LLM training and theoretical bounds on the loss at the final iterate of SGD in a non-smooth stochastic convex setting. Through empirical studies on Llama-style transformers, the paper demonstrates that the theoretical bound effectively predicts the actual loss curve based on the learning rate schedule. Building on this observation, the authors show how insights from these theoretical bounds can guide learning-rate tuning for continued training and enable learning-rate transfer across different schedules in practice.

## update after rebuttal

After carefully considering the authors’ responses in their rebuttal, my assessment remains unchanged, and I continue to recommend acceptance. The authors have adequately addressed my initial concerns, and the clarification provided further strengthens the contributions of the paper.

**Claims And Evidence:**

The main claim of the paper is that the theoretical performance bound on the final iterate of SGD in non-smooth stochastic convex optimization aligns well with the loss curve in LLM training. This claim is strongly supported by a series of experiments on language model training, as well as theoretical simulations.

**Essential References Not Discussed:**

The paper discusses all the essential references related to the main contribution.

**Experimental Designs Or Analyses:**

Overall, the experiments are well-designed, and the results look convincing.

**Methods And Evaluation Criteria:**

The proposed application of the theoretical bound—schedule construction for continued training and learning-rate transfer across schedules—seems convincing and could be a valuable tool in practice.

**Other Comments Or Suggestions:**

Here are some additional experiments I would suggest:
1. Experiments using SGD on vision tasks could provide insights into the broader applicability of the theoretical bound.
2. Testing a more diverse range of learning rate schedules beyond the popular cosine and WSD, and comparing them with the theoretical bound, would be interesting. For example, in Tissue et al. (2024), the authors tested various learning rate schemes—see Figures 3 and 4.

However, the current paper remains valuable even without these additional experiments.

Tissue et al. (2024) - Scaling Law with Learning Rate https://arxiv.org/abs/2408.11029

**Other Strengths And Weaknesses:**

Overall, this paper is well-written and makes a solid contribution by connecting non-smooth convex theory with language model training. This bridges a critical gap between optimization theory and deep learning practice, which I find to be very interesting.

The main strength lies in demonstrating that the theoretical performance bound has strong predictive power in practice.

There are a few minor weaknesses. First, while the non-smooth convex theory is derived from SGD, all the experiments in this paper use AdamW. This limitation is well acknowledged in the paper. The paper would be even stronger if it included experiments using SGD and compared the loss curves with the theoretical bound.

**Questions For Authors:**

1. Does the alignment between non-smooth convex optimization theory and learning rate scheduling apply only to “large” model training?
2. Does the alignment hold only for AdamW, or can similar observations be made for other modern optimizers (e.g., Shampoo, Schedule-Free, Muon, MARS, etc.)?

**Relation To Broader Scientific Literature:**

N/A

**Theoretical Claims:**

The main theoretical framework is adopted from the existing work by Defazio et al. (2023). Building on these theoretical results, this paper derives the theoretical bound for different learning rate scheduling scenarios. These theoretical results appear to be correct.

---

> ### Author Rebuttal · Authors · 2025-03-28
>
> We thank the reviewer for their positive feedback, especially for the assessment that our experimental results are strongly supporting the claims/contributions.
>
> * We ran additional experiments with SGD on Imagenet as suggested by the reviewer (see details here: https://anonymous.4open.science/r/icml25-additional-experiments-lr-schedules/). The results confirm the close match between the bound and actual loss curves.
> * Regarding additional schedules, we would like to point to Fig 18 and 23 where we compare many different schedules beyond cosine and WSD, including constant and 1/sqrt schedule as well as the 1-sqrt schedule proposed in Hägele et al 2024. We also test a piecewise constant schedule with cooldown for the continual learning experiment.
> * Regarding other optimizers, we have not done a systematic comparison. However, from results reported in the literature, it is known that the sudden drop of the loss also appears for other optimizers, for example Muon, Shampoo and SOAP (see https://github.com/KellerJordan/modded-nanogpt/tree/master/records/102924_Optimizers) or Ademamix (see Fig 3b in https://arxiv.org/pdf/2409.03137). This makes us confident that the characteristic schedule behaviours that we describe also generalize to other optimizers; investigating this in detail would be an interesting direction for followup work.

---

> > ### Comment · Reviewer_hqnc · 2025-04-04
> >
> > Thank you for the detailed response and for conducting the additional experiments. I have a follow-up question regarding the SGD results on ImageNet. In Figure 25, the theoretical bound does not appear to align closely with the validation loss curve, which is understandable given the multi-pass setting. However, in Figure 26, the training loss curve also seems to deviate from the theoretical bound. For example, the predicted loss at the final iterate based on the theoretical bound does not match the actual training loss in terms of the relative ordering across different schedulers. Could the authors clarify this discrepancy?

---

> > > ### Author Response · Authors · 2025-04-04
> > >
> > > Dear reviewer,
> > >
> > > thank you for your comments.
> > >
> > > For the theoretical plot (Fig 25a), we had used the same three base LRs for cosine and wsd, whereas the empirical plots shows a range of base LRs which are doubled for cosine (in line with our findings that cosine needs roughly a twice as large LR). The updated theoretical plot (same link as before) now matches the empirical plot also in terms of relative ordering - **thanks for catching this!**
> > > Another minor factor is the range of LRs, which we slightly reduced (the sensitivity to base LR depends on problem constants, which are unknown and can be different for datasets, models etc).
> > >
> > > Two additional remarks on the theory-practice alignment for Imagenet: 1) in our experience the randomness on Imagenet is slightly larger (might be due to data augmentation), and we had only ran one seed due to the limited rebuttal time. Some differences might be only due to this noise. 2) We noticed that for Imagenet the gradient norms are slowly increasing over training, in contrast to the LLM training, where the gradient norm is almost constant (see e.g. also Fig 3 in https://arxiv.org/pdf/2310.07831). We think that the impact of gradient norms on the performance of schedules is an interesting direction for follow-up work.

---

### Official Review · Reviewer_rpAe · 2025-03-13

**Overall Recommendation:** 3

**Summary:**

This work demonstrates that several empirical observations align with the last-iterate sub-optimality gap in convex optimization. Furthermore, the authors show that adjusting the learning rate in continual learning—an approach that theoretically improves this sub-optimality gap—also enhances real-world training performance. Lastly, the study establishes a convergence guarantee for the WSD schedule.


## update after rebuttal

Thank you for your detailed response.

In light of the additional experiments, I will raise my score.

**Claims And Evidence:**

The claims linking the theoretical bounds to practical performance are supported by the experiments in the paper. However, the experiments are limited to a single synthetic problem and one real dataset. Consequently, the assertion that the theoretical bounds accurately reflect real-world behavior—allowing for extrapolation from the sub-optimality gap—is not strongly justified.

**Essential References Not Discussed:**

The work of Liu and Zhou establishes a last-iterate convergence guarantee that applies to both convex Lipschitz and convex smooth problems, including mirror descent. These results can also be leveraged to obtain guarantees in the convex smooth case. Notably, (A) these guarantees rely on Lipschitz/variance bounds rather than the $G_t$ assumption used in this paper, and (B) the constants in this work may be slightly worse.

Z. Liu and Z. Zhou. Revisiting the last-iterate convergence of stochastic gradient methods. In The Twelfth International Conference on Learning Representations, 2024.

**Experimental Designs Or Analyses:**

In general, the experimental design and analysis appear valid. The behavior of WSD across the experiments aligns with findings from previous work.

**Methods And Evaluation Criteria:**

The methods and evaluation criteria are well-founded. However, experiments on additional datasets would further strengthen the analysis.

**Other Comments Or Suggestions:**

No additional comments.

**Other Strengths And Weaknesses:**

**Strengths**
- The work presents multiple observations that align well between theory and experiments.

**Weaknesses**
- The theoretical results are relatively straightforward. While the reviewer is not aware of prior results specifically for the WSD schedule, deriving such guarantees using existing general last-iterate convergence results does not require significant innovation.
- Additional experiments would substantially strengthen this work.

**Questions For Authors:**

- Given the last-iterate guarantee for convex smooth objectives established by Liu and Zhou (referenced above), should there be any notable differences in the predictions and alignment of this work, which assumes convex Lipschitz objectives? This question assumes that the number of steps is sufficiently large for the $1/T$ smooth term to be negligible.
- Can the authors provide justifications for Assumption A3 beyond a general Lipschitz assumption? While the reviewer is familiar with optimization theory, they are not aware of other instances where this assumption has been used, especially since it depends on the algorithm, as the authors mentioned.

Overall, while this work presents interesting observations, its limited empirical evaluation and lack of significant theoretical novelty make it a borderline case for acceptance.

**Relation To Broader Scientific Literature:**

Understanding the relationship between theory and practice is an important topic for the machine learning community.

**Theoretical Claims:**

Given the last-iterate guarantee from previous work, which accommodates general step-size sequences, the convergence result is reasonable and unsurprising. It is well established that a linearly decaying step-size sequence leads to last-iterate convergence guarantees. Therefore, when the cooldown period is of the same order as the training duration, the observed behavior aligns with the theoretical expectations.

---

> ### Author Rebuttal · Authors · 2025-03-28
>
> We thank the reviewer for their thoughtful review and detailed feedback. We address all questions in detail below. We hope that this will clarify all concerns, and allows for a higher scoring of our submission.
>
> * *limited empirical evaluation, one real dataset:* We agree that it is beneficial to validate our findings on additional datasets, and for the rebuttal we performed additional experiments on Imagenet and OpenWebText; the results confirm our previous findings (see detail here https://anonymous.4open.science/r/icml25-additional-experiments-lr-schedules/). However, even though we had only considered one dataset, our empirical evaluation is not limited: we verify our findings on multiple schedules+model sizes+time horizons, and **find that multiple aspects of the theory match the experiments** (e.g. schedule adaptation for longer runs, effect of cooldown length, LR transfer across cooldown lengths etc.).
> Besides the dataset choice (which we address in the rebuttal), are there any other reasons why the reviewer considers our empirical evaluation to be limited?
> * Related to the above, the characteristic behaviour of WSD during cooldown has been reported in multiple papers (which all train in slightly different settings and on different datasets). For example (see also references in our paper): https://arxiv.org/pdf/2410.05192, https://arxiv.org/abs/2502.15938.
> These independent experiments further justify that the bound indeed reflects real-world behaviour, and our paper is the first one to highlight and exploit this connection between theory and practice.
>
> * *lack of significant theoretical novelty*: One interesting theoretical insight is that the cooldown of WSD achieves to remove a log-factor in the convergence bound (compared to the constant schedule). Moreover, this improvement is obtained exactly by the drop during cooldown, as Figure 20 (left) shows. More broadly, the main goal of our paper is *not* to develop new proof techniques, but to *show how existing theory can be used* in order to (i) design/tune the learning rate (schedule) and (ii) explain the benefit of cooldown observed in practice (and the behaviour of LR-schedules in general).
>
> * Correct, Assumption A3 is implied when assuming Lipschitzness of the objective. However, as the bound only depends on the expected gradient norms on the SGD trajectory (and the proof does not explicitly use the Lipschitz condition), we wanted to formulate the assumption as tight as possible - bounded gradient norms over a finite number of steps could be satisfied with much weaker assumptions than Lipschitz continuity.
>
> * We thank the reviewer for pointing out the work by Liu and Zhou; we were not aware of this paper and will discuss it in the related work section. From our understanding the main differences are:
>   - the results by Liu and Zhou do not hold for arbitrary schedules
>   - In the Lipschitz-smooth case (M=0 in Liu and Zhou), instead of the Lipschitz constant, we have the noise constant in the bound, and additionally the step size needs to be smaller than $\mathcal{O}(1/2L)$. However, as we can decompose the bound $G_t$ in our notation in a noise part plus the Lipschitz constant of the objective $f$, this appears to result in a very similar structure.
> Extending our analysis to the smooth case is definitely interesting and we are actually working actively in this direction.

---

> > ### Comment · Reviewer_rpAe · 2025-04-02
> >
> > Thank you for your detailed response.
> >
> > In light of the additional experiments, I will raise my score.
> >
> > For the benefits of the authors, I will also mention that Liu and Zhou in fact do prove convergence guarantees for arbitrary schedules, see in Lemma 4.2, in case the authors would like to extend their results to the smooth case. (I do not ask for this addition, only mentioning it for the benefits of the authors.)

---

> > > ### Author Response · Authors · 2025-04-03
> > >
> > > We thank the reviewer again for their efforts to review at ICML, and are delighted that our response and new experiments lead to a higher score.

---

### Decision · Program_Chairs · 2025-05-01

**Decision:**

Accept (poster)

**Comment:**

This paper provides an analysis of the WSD schedule in stochastic convex optimization by leveraging some general theorems recently developed for understanding learning rate schedules in this setting. The authors also extract a particular anytime convergence result from prior work, and conduct an empirical investigation suggesting that this (convex) theory appears to have good predictive power for some (non-convex) transformer training tasks. Overall, the empirical results are perhaps of too small a scale to provide confidence in modern practical settings. However, even at this scale, optimization theory is notoriously poor at predicting empirical results, so this study may be of interest to the optimization theory community.